# Filtration effect of *Cordyceps chanhua* mycoderm on bacteria and its transport function on nitrogen

Gongping Hu,[1] Yeming Zhou,[1] Dan Mou,[2] Jiaojiao Qu,[1,3] Li Luo,[1] Lin Duan,[1] Zhongshun Xu,[1] Xiao Zou[1]

**ABSTRACT** *Cordyceps chanhua* is a kind of entomopathogenic fungus that has a long medicinal history in traditional Chinese medicine. In this study, we aimed to explore the role of *C. chanhua* mycoderm. Our results showed that *C. chanhua* mycoderm has a certain ability to transfer nitrogen, which can transfer nitrogen from the insect body to the external environment and transfer nitrogen from the environment to *C. chanhua*. By studying the filtration of bacterial mycoderm, it was found that the mycoderm of *C. chanhua* was able to filter out most of the bacteria in the environment. Using the high-throughput sequencing methods, we found that the abundance of bacterial community first increased and then decreased during the growth and development of *C. chanhua*. The bacterial richness and diversity of mycoderm of *C. chanhua* cultivated in sterile glass bottles were significantly lower than those cultivated in soil mulching. In addition, this study also found that the mycoderm of *C. chanhua* formed under both soil mulching cultivation and sterile glass bottle cultivation was hydrophobic. The mycoderm of *C. chanhua*, which occurred under sterile glass bottle cultivation, was closely and orderly arranged, while the mycoderm of *C. chanhua* cultivated in soil was quite the opposite. The density and thickness of the *C. chanhua* membrane in aseptic glass bottle culture were higher than those in soil-covered culture, but the dry/fresh weight was lower than those in soil-covered culture.

**IMPORTANCE** During the natural growth of *Cordyceps chanhua*, it will form a mycoderm structure specialized from hyphae. We found that the bacterial membrane of *C. chanhua* not only filters environmental bacteria but also absorbs and transports nitrogen elements inside and outside the body of *C. chanhua*. These findings are of great significance for understanding the stable mechanism of the internal microbial community maintained by *C. chanhua* and how *C. chanhua* maintains its own nutritional balance. In addition, this study also enriched our understanding of the differences in bacterial community composition and related bacterial community functions of *C. chanhua* at different growth stages, which is of great value for understanding the environmental adaptation mechanism, the element distribution network, and the changing process of symbiotic microbial system after *Cordyceps* fungi infected the host. At the same time, it can also provide a theoretical basis for some important ecological imitation cultivation technology of *Cordyceps* fungi.

**KEYWORDS** *Cordyceps chanhua*, bacterial community, mycoderm morphology, nitrogen transport, selective filtration

Address correspondence to Xiao Zou, xzou@gzu.edu.cn.

The authors declare no conflict of interest.

See the funding table on p. 19.

Cordyceps chanhua is a parasitic complex fungus formed by *Cordyceps* fungi parasitizing cicada insects (1). *C. chanhua* is composed of five parts, which are coremium, spore powder, mycoderm, insect body wall, and inner sclerotia (Fig. 1), and its growth and development are closely related to its habitat (2, 3). *C. chanhua* was recorded for the first time in the *Lei Gong's Theory of Blazing* written by Lei Xiao in the

**FIG 1** Schematic diagram of the structure of the *C. chanhua*. (A) Soil-covered cultivated *C. chanhua*. (B) Sterile glass bottle cultivated *C. chanhua*.

Northern and Southern Dynasties, 800 years earlier than *Cordyceps sinensis* (4, 5). Chinese scholar Li Zengzhi found the sexual *C. chanhua*, classified it as a new species of the genus *Cordyceps*, and named it *Cordyceps chanhua* with the ancient name of *C. chanhua* (6). As a traditional Chinese herbal medicine, *C. chanhua* has many effective ingredients such as nucleoside, ergosterol, polysaccharide, cordyceps acid, polyglobulin, amino acid, and fatty acid (7–9). In terms of medicinal value, *C. chanhua* has antibacterial and antioxidant properties (10), anti-tumor properties (11), enhanced immune regulation (12), protected and improved renal function (13, 14), anti-aging (15), anti-fatigue (16), hypoglycemic (17), and other pharmacological effects. As an entomopathogenic fungus, *C. chanhua* is pathogenic to a variety of agricultural pests, including Lepidoptera, Homoptera, and Hemiptera, such as *Plutella xylostella*, *Trialeurodes vaporariorum*, *Coptotermes formosanus*, and aphid (18–20). In addition, *C. chanhua* is also widely used in daily life; an interesting study reported that *C. chanhua* has a high flocculation effect on coal washing wastewater in early research (21).

The artificial cultivation techniques of *C. chanhua* include liquid fermentation, solid culture, insect body culture, and ecological mulching soil culture (22, 23). Among these methods, the insect body part and fruiting body part of *C. chanhua* cultivated in insect body culture and ecological mulching soil culture can be harvested and have similar appearance characteristics to wild *C. chanhua*, which make them more popular (2, 24). Our previous research found that the contents of several major active substances, such as cordycepin and adenosine in the bionic soil-covered *C. chanhua*, were similar to those in the wild *C. chanhua* (3). Therefore, from the perspective of active ingredients, the soil-covered *C. chanhua* can be substituted for the wild *C. chanhua*.

At present, most studies on the morphological characteristics of *Cordyceps* fungi focus on the hypha, sclerotium, stroma, and ascus structures of cordyceps (25–27). However, as the contact surface between *C. chanhua* and the external environment, the morphological characteristics of the mycoderm has not been reported. Previous studies have found that the relative abundance of other fungi in the sclerotia of *C. chanhua* was relatively low while the bacterial community was relatively rich (1, 3, 28). The mycoderm is the interface between *C. chanhua* and the environment, while the microbial community richness of entomopathogenic fungal mycoderm is significantly higher than that of inner

sclerotia and bacteriosphere soil (29, 30). Therefore, exploring the biofiltration effect of *C. chanhua* membrane on a bacterial community is a specific priority in this study.

In this study, we analyzed the bacterial community of the inner sclerotium and mycoderm of *C. chanhua* at different growth stages and under different cultivation environments by high-throughput sequencing. We also analyzed the filtration effect of *C. chanhua* membrane on bacterial communities. In addition, we also analyzed the nitrogen transport function of the membrane of *C. chanhua*.

## RESULTS

### Bacterial microbial diversity in sclerotia and mycoderm of *C. chanhua* at different growth stages

#### *Alpha diversity analysis*

The dilution curve analysis found that the Shannon-Wiener curve in this study tends to be flat with the increase of the number of sample sequences, indicating that the amount of sequencing data is reasonable and the sequencing depth can cover most of the microbial diversity information in the sample bacterial community (Fig. S1). After optimization, 2,112,984 valid sequences were detected in 27 samples of this sequencing, with a total of 898,572,190 bases. Each sample was 40,387–90,034, with an average sequence length of 425 bp. The optimized sequence was clustered by operational taxonomic unit(OTU) with 97% consistency. A total of 3,850 OTUs of 1,785 species, 953 genera, 429 families, 41 phyla, 101 classes, and 202 orders were detected in nine groups of samples of sclerotia and mycoderm in *C. chanhua* in different cultivation environments and different growth stages.

The $a$ diversity index of different samples was compared and analyzed, and the coverage index of each sample was more than 99%, pointing that the data measured by this sequencing can adequately reflect the diversity of the bacterial community in the sclerotia and the mycelium of *C. chanhua* (Table 1). The Ace and Chao indexes in sclerotia of *C. chanhua* in different cultivation periods wereu 90,059–414,584 and 66,074–409,652, respectively. There was no significant difference between sclerotia bacterial richness of *C. chanhua* cultivated in sterile glass bottles and covered with soil in different growth stages ($P > 0.05$); the results showed that the bacterial community structure of sclerotium samples in *C. chanhua* in different growth stages was similar. The Ace and Chao indexes of sclerotia in the two stages of *C. chanhua* covered with soil were 338,996–414,584 and 299,156–409,652, respectively, and the richness was significantly higher than that in the sclerotia samples of *C. chanhua* under aseptic cultivation ($P < 0.05$). It can be seen from Table 1 that the richness and diversity of *C. chanhua*

**TABLE 1** Statistics of alpha diversity of bacterial community in each sample[a]

| Sample | Ace index | Chao index | Coverage (%) | Shannon index | Simpson index |
|---|---|---|---|---|---|
| Sclerotia | | | | | |
| Si_1 | 134.129 ± 28.019ab | 132.836 ± 28.810ab | 99.961 ± 0.013a | 1.019 ± 0.082a | 0.496 ± 0.054a |
| Si_2 | 157.259 ± 157.319ab | 154.487 ± 160.912ab | 99.834 ± 0.236a | 0.832 ± 0.494a | 0.631 ± 0.250ab |
| Si_3 | 90.059 ± 45.404b | 66.074 ± 24.839b | 99.942 ± 0.060a | 0.523 ± 0.526a | 0.772 ± 0.271b |
| Ss_2 | 414.584 ± 118.020c | 409.652 ± 113.673c | 99.961 ± 0.027b | 1.015 ± 0.336a | 0.644 ± 0.168ab |
| Ss_3 | 338.996 ± 371.656bc | 299.156 ± 301.112bc | 99.879 ± 0.069ab | 1.068 ± 1.354a | 0.710 ± 0.337ab |
| Mycelium cortices | | | | | |
| Ci_2 | 96.201 ± 4.044a | 87.600 ± 7.208a | 99.961 ± 0.004a | 1.131 ± 0.321a | 0.537 ± 0.173a |
| Ci_3 | 244.316 ± 241.623a | 219.579 ± 262.342a | 99.958 ± 0.012a | 1.590 ± 1.012a | 0.396 ± 0.109a |
| Cs_2 | 2,053.373 ± 111.635b | 2,043.340 ± 115.266b | 99.249 ± 0.035b | 5.595 ± 0.130b | 0.012 ± 0.002b |
| Cs_3 | 1,441.984 ± 139.767c | 1,407.231 ± 150.996c | 99.372 ± 0.174b | 4.702 ± 0.537b | 0.028 ± 0.014b |

[a]The S is expressed as the sclerotia, and C represents a mycelium cortices. Si_1: *C. chanhua* sclerotia cultivated in sterile glass bottle at stage of ossified cicada; Si_2: *C. chanhua* sclerotia cultivated in sterile glass bottle at cortical formation stage; Ci_2 represents its mycelium cortices; Si_3: *C. chanhua* sclerotia cultivated in sterile glass bottle at mature period; Ci_3 represents its mycelium cortices; Ss_2: *C. chanhua* sclerotia cultivated in soil at mycelium cortical formation stage; Cs_2 represents its mycelium cortices; Ss_3: *C. chanhua* sclerotia cultivated in soil at mature period; Cs_3 represents mycelium cortices. Different lowercase letters represent statistically significant differences ($α = 0.05$).

mycoderm under soil mulching cultivation are significantly higher than those under aseptic cultivation and were conducted to prove that different cultivation environments can affect the composition and diversity of bacterial community of *C. chanhua* mycoderm. The bacterial community richness of soil-covered *C. chanhua* in the mycoderm forming stage was significantly higher than that of mature *C. chanhua* and showed that the number of bacterial species in the mycoderm forming stage was large. In each mycoderm sample, at the phyla level, the bacterial community composition of *C. chanhua* mycoderm in each stage of soil-covered cultivation and sterile glass bottle cultivation is similar (Fig. S2A). At the genus level, the composition of bacterial communities in the film of *C. chanhua* under soil mulching cultivation is relatively complex, and the composition of bacterial communities in the film formation stage and mature stage is similar (Fig. S2B).

### Beta diversity analysis

The distance between Si_1 and other internal sclerotium samples is relatively far, shown in Fig. 2, which indicated that the bacterial community composition of sclerotia samples in the zombie stage was significantly different from that in other stages of *C. chanhua*. The sclerotia (Si_2 and Si_3) and mycoderm samples (Ci_2 and Ci_3) in aseptic culture *C. chanhua* were gathered separately, while the sclerotia samples in soil-covered culture *C. chanhua* were scattered in each stage (Ss_2 and Ss_3); these results indicated that the composition of sclerotium bacterial community in each stage was significantly different while the composition of mycoderm bacterial community in each stage was relatively similar. The distance between the samples of aseptic cultivation and the samples of soil cultivation indicates that the composition of bacterial community between the two groups was quite different.

### OTUs shared by sample communities at different cultivation stages in the same environment

At the genus level, the sclerotium and mycoderm samples of *C. chanhua* cultivated in aseptic glass bottles in different cultivation stages were analyzed by Venn diagram. The results showed that 264 OTUs were detected at the stage of *C. chanhua* fossilization (Si_1); 552 OTUs were detected in sclerotia and 87 OTUs were detected in mycoderm formation stage; and 170 OTUs were detected in sclerotia and 534 OTUs in mycoderm during *C. chanhua* maturation (Fig. 3A). The analysis found that there were 47 OTUs in sclerotia and 35 OTUs in mycoderm during the three stages of *C. chanhua*; the results demonstrated that some bacteria always accompanied the growth and development of *C. chanhua*. There were more sclerotium bacteria in *C. chanhua* than in zombie stage during the mycoderm formation stage, and there were fewer bacteria in the mycoderm at this time; with the extension of cultivation time, the number of bacteria in sclerotia decreased gradually, while the number of bacteria in mycoderm increased significantly. It can be inferred that with the growth and development of *C. chanhua* and the formation of bacterial mycoderm, some bacteria may migrate into the inner sclerotia.

A total of 1,142 OTUs were detected in sclerotia, and 2,097 OTUs were detected in sclerotia of *C. chanhua* cultivated with mulching soil during the period of mycoderm formation; 942 OTUs were detected in sclerotia and 1,619 OTUs in mycoderm during *C. chanhua* maturation. On the basis of these results, we can find that the number of bacteria in sclerotia and mycoderm in *C. chanhua* significantly increased and then decreased due to the complexity of soil environment after the soil covering cultivation of the stiff insects (Fig. 3B). There are 86 OTUs between sclerotia and 819 OTUs in the mycoderm during the growth of *C. chanhua*, which indicates that some bacteria in sclerotia are retained during the stage of rigor mortis and participate in the whole process of *C. chanhua* growth and development.

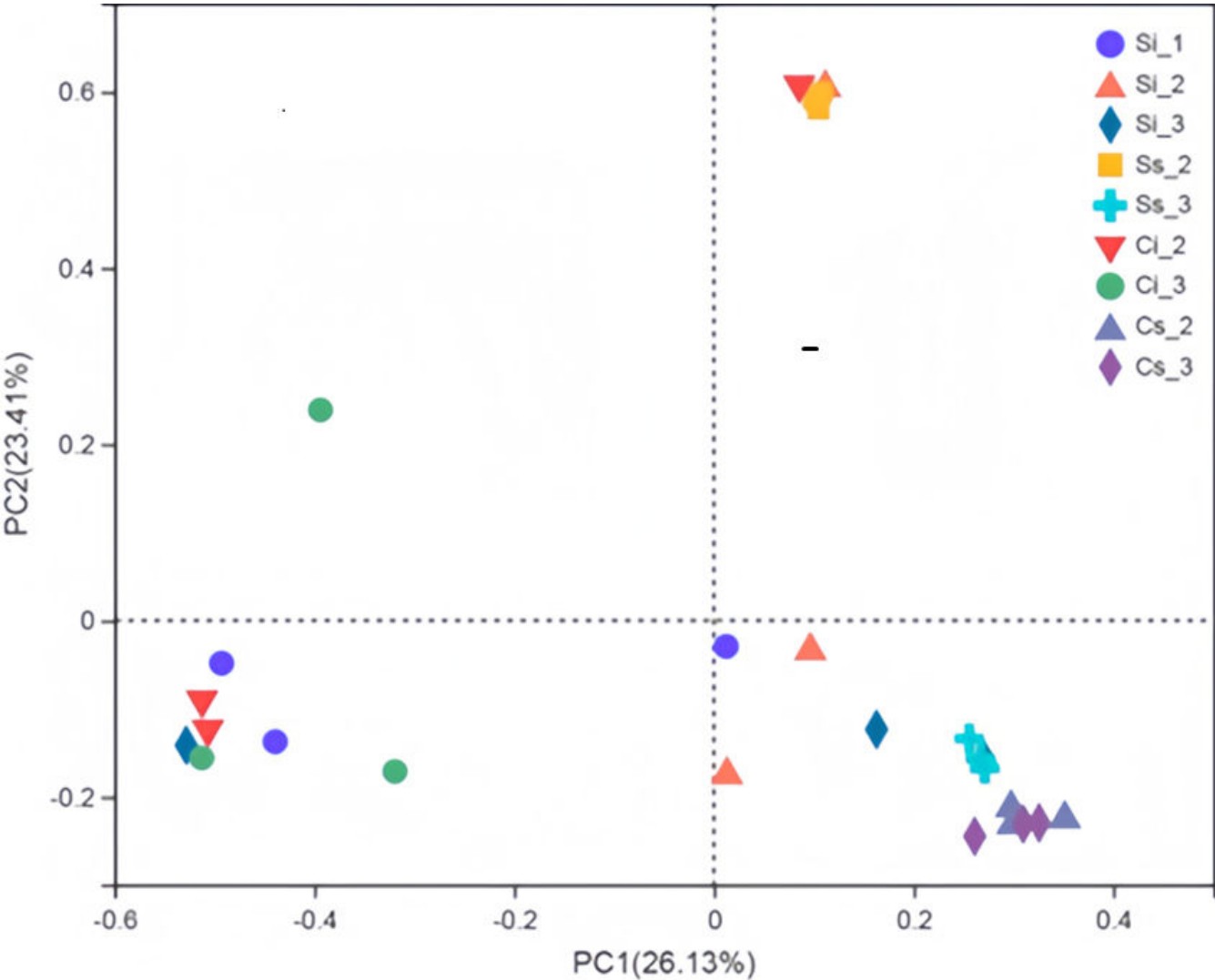

**FIG 2** PCoA analysis of bacterial communities. Note: The S is expressed as the Sclerotia, and C represents a mycelium cortices. Si_1: *C. chanhua* sclerotia cultivated in sterile glass bottle at stage of ossified cicada; Si_2: *C. chanhua* sclerotia cultivated in sterile glass bottle at cortices formation stage, Ci_2 represents its mycelium cortices; Si_3: *C. chanhua* sclerotia cultivated in sterile glass bottle at mature period, Ci_3 represents its mycelium cortices. Ss_2: *C. chanhua* sclerotia cultivated in soil at mycelium cortices formation stage, Cs_2 represents its mycelium cortices; Ss_3: *C. chanhua* sclerotia cultivated in soil at mature period, Cs_3 represents mycelium cortices.

### OTUs shared by bacterial communities in different cultivation environments at the same stage

Venn analysis was carried out on sclerotia and mycoderm of *C. chanhua* in different cultivation environments at the same stage. During the mycoderm formation period, 552 OTUs were detected in sclerotia, and 87 OTUs were detected in the sterile-cultivated *C. chanhua*; there were 1,142 OTUs for sclerotia and 2,097 OTUs for mycoderm in *C. chanhua* cultivated with mulching soil. There are 211 OTUs in the inner sclerotia and 48 OTUs in the mycoderm of the two environments, indicating that the bacterial community composition is different in different *C. chanhua* cultivation environments (Fig. 4A). At the mature stage of *C. chanhua*, 170 OTUs were detected in the sclerotia and 543 OTUs in the bacterial mycoderm of sterile *C. chanhua*. There were 942 OTUs for sclerotia and 1,619 OTUs for mycoderm in *C. chanhua* under mulching cultivation. There are 67 OTUs between inner sclerotia and 98 OTUs between mycoderms, which indicates that the

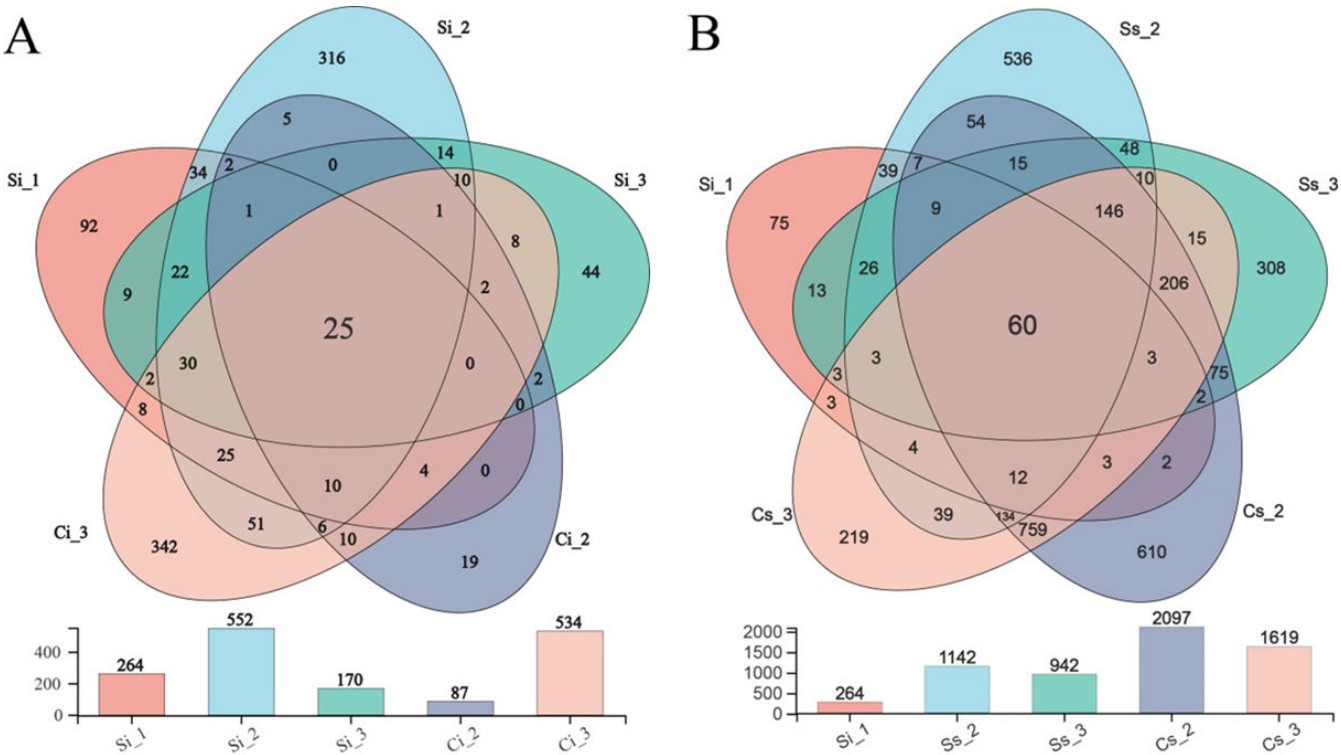

**FIG 3** Venn diagram analysis of bacterial communities in *C. chanhua* at different cultivation stages with OTU level. (A) *C. chanhua* cultivated in aseptic glass bottle. (B) *C. chanhua* cultivated in soil.

longer the growth time is in different environments, the greater the difference in the composition of bacterial community structure of *C. chanhua* (Fig. 4B).

### Common genera of bacterial community in C. chanhua samples

At the genus level, a petal diagram was drawn for all sample bacterial communities of *C. chanhua* in different stages of the two cultivation environments. The results showed that there were 22 genera in all sample bacteria (Fig. 5A), and the pie-shaped distribution diagram of their proportions was shown in Fig. 5B. They are *Achromobacter* (35.42%), *Serratia* (24.90%), *Pseudomonas* (19.33%), *Cedecea* (6.79%), *Enterococcus* (5.70%), *Bacillus* (2.49%), *Streptomyces* (1.66%), *Pantoea* (0.90%), and other 14 small bacterial groups. These bacteria were detected in all samples; the results demonstrated that they participated in the whole process of *C. chanhua* growth and development.

As shown in Fig. 5A, the sample Si_1 has 12 endemic genera, which are unclassified Lactobacillales, *Spongiimonas*, *Tetragenococcus*, etc. The sample Si_2 has endemic genera, which are *Aequoorivita*, *Marinilactibacillus*, and other 20 genera; Si_3 endemic genera are *Cytophaga*, *Histophilus*, *Oceanivirga*, and *Chitinimonas*; Ci_2 endemic genera are *Paenochrobactrum* and *Nesterenkonia*; and Ci_3 endemic genera are *Prevotella*, *Gelria*, *Ochrobactrum*, *Pirellula*, and other 59 genera. There are many endemic genera of *C. chanhua* under mulching cultivation. Ss_2 endemic genera are *Candidatus*, *Rhabdochlamydia*, and other 82 genera; Ss_3 endemic genera include 50 genera, such as *Proteiniclasticum* and *Desulfomonile*; Cs_2 endemic genera include 26 genera, which are *Thermopolyspora*, *Litorilina*, etc.; Cs_3 endemic genera include 16 genera, which are *Dyadobacter*, *Mycoavidus*, and *Leadbetterella*. In addition to 22 bacterial genera participating in the growth and development of *C. chanhua* in all samples, *C. chanhua* samples in different environments and stages have their own unique bacterial genera participating in the development of specific periods.

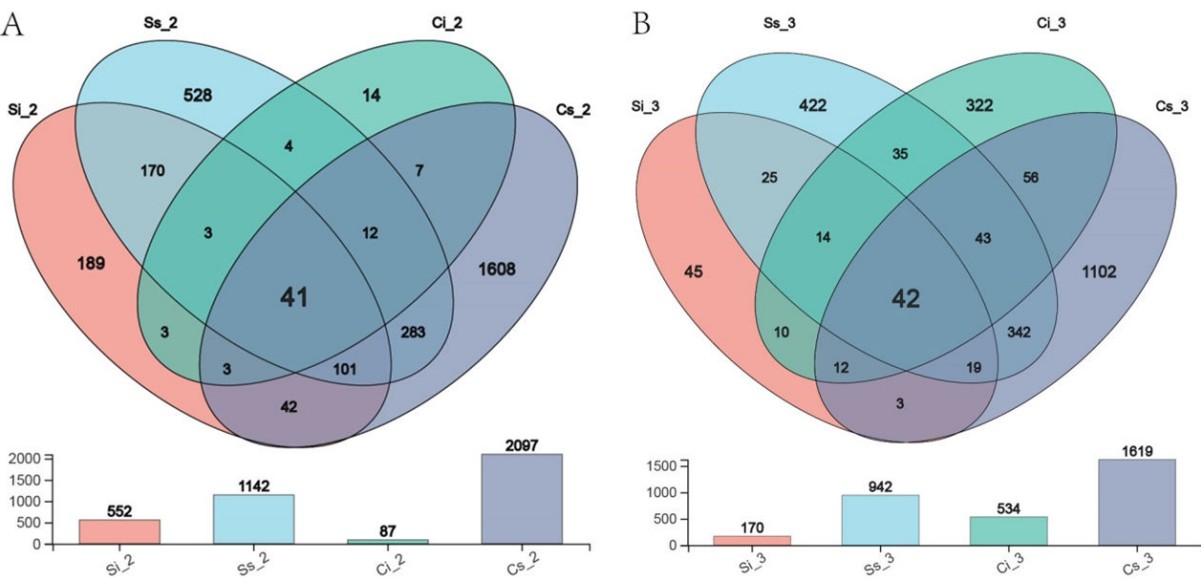

**FIG 4** The common analysis of sclerotium and plaque bacterial communities in *C. chanhua* in different cultivation environments at the same level of OTU classification. (A) *C. chanhua* at mycelium cortical formation stage. (B) *C. chanhua* at mature period.

## Analysis of differences between groups

The results of the significant difference test between groups showed that there were significant differences among the top 15 bacteria in *C. chanhua* abundance in different cultivation environments and stages (Fig. 6), including *Achromobacter*, *Pseudomonas*, *Enterococcus*, *Bacillus*, *Morganella*, *Candidatus_ Rhabdochlamydia*, *Mycobacterium*, *Bradyrhizobium*, *Acidothermus*, etc. The abundance of *Achromobacter* was higher in the later period of cultivation, which was significantly different from that in the rigid stage. However, the abundance of *Enterococcus* and *Bacillus* was higher in the zombie stage, and the abundance of the later cultivated samples decreased or disappeared significantly in the middle stage; *Mycobacterium*, *Bradyrhizobium*, and *Acidothermus* almost only appeared in the two stages of mulching cultivation, which was significantly different from sterile cultivation. In different cultivation environments, the bacterial communities

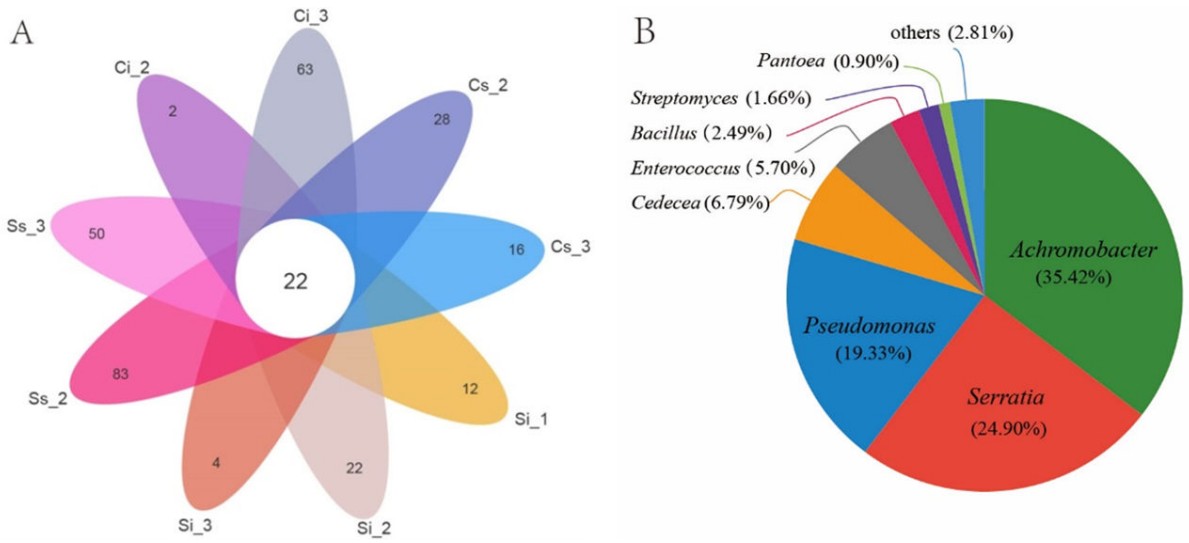

**FIG 5** Common genera of bacterial community in *C. chanhua* samples. (A) Petal diagram of bacterial communities in each sample at the genus classification level. (B) Pie-shaped distribution diagram of the proportion of 22 common genera.

at different stages of *C. chanhua* are different, and their functions are also different (Fig. S3 and S4).

## Apparent morphology of *C. chanhua* mycoderm in soil and atmosphere

### Hydrophobicity

The water contact angle of both sterile glass bottle and soil-covered *C. chanhua* mycoderm is greater than 90° (Table 2), indicating that the mycoderm in both environments is hydrophobic. The hydrophobicity of *C. chanhua* cultured in sterile glass bottle is significantly stronger than that of the soil-covered *C. chanhua* mycoderm, indicating

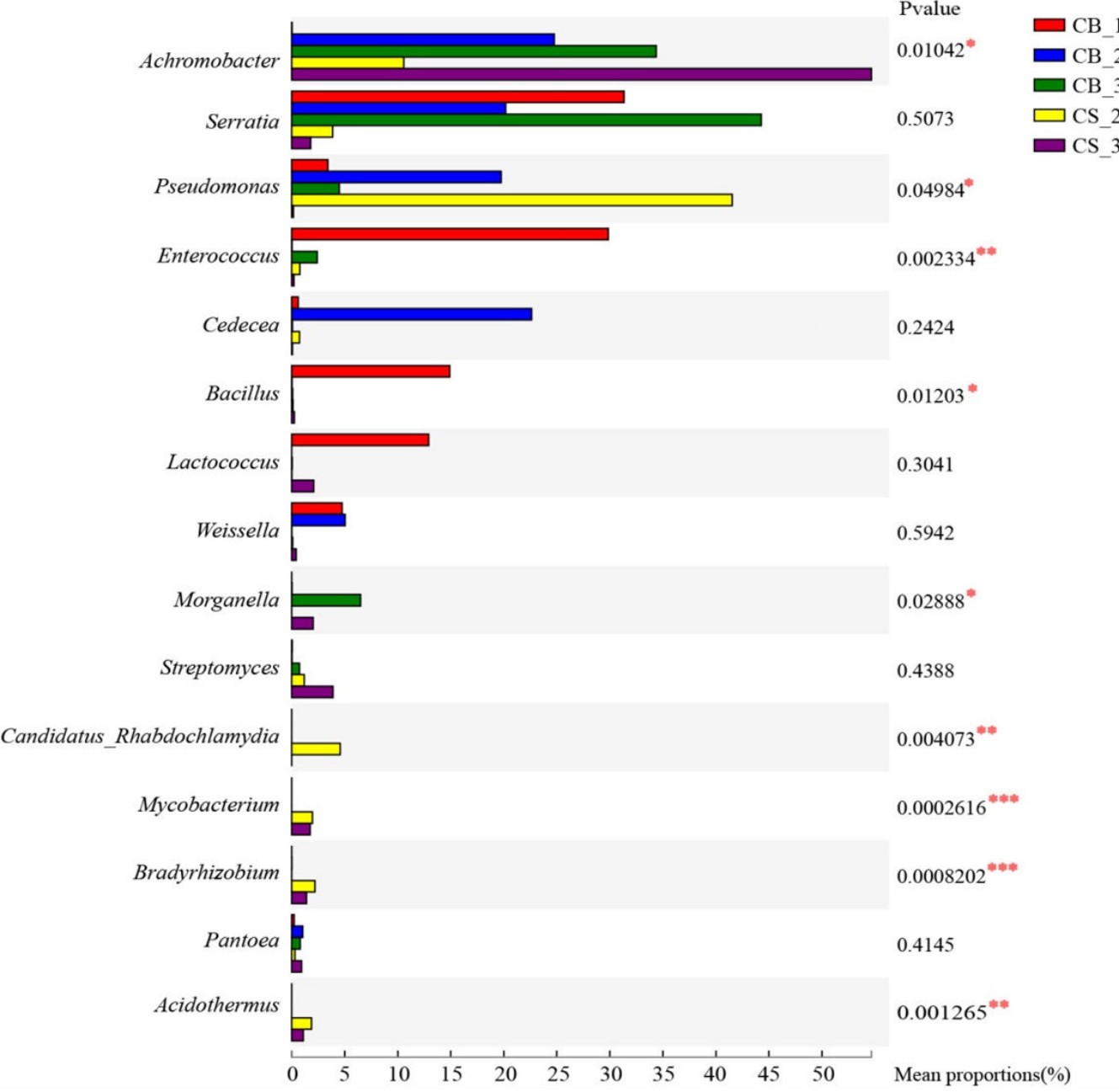

**FIG 6** Analysis of significant differences between each sample of artificial cultivation *C. chanhua*. (CB_1) *C. chanhua* in the ossified cicada stage. (CB_2) *C. chanhua* in the sterile glass bottle at mycelium cortical formation stage. (CB_3) *C. chanhua* in the mature period of sterile glass bottle cultivation. (CS_2) *C. chanhua* in the soil at mycelium cortical formation stage. (CS_3) *C. chanhua* in the soil at mature period.

**TABLE 2** Water contact angle (θw) of *C. chanhua* mycelium cortices in different cultivation environments[a]

| Mycelium cortices | Water contact angle | Fungal surface classification |
|---|---|---|
| Glass bottle cultivated *C. chanhua* | 123.3°±2.5a | Hydrophobic |
| Soil-covered cultivated *C. chanhua* | 101.4°±1.6b | Hydrophobic |

[a]Different lowercase letters represent statistically significant differences ($a = 0.05$).

that *C. chanhua* can adjust its hydrophobicity strength to adapt to the environment in different environments.

### Weight and thickness of mycoderm

The fresh/dry weight of *C. chanhua* mycoderm cultivated with soil mulching is significantly higher than that of *C. chanhua* mycoderm cultivated in sterile glass bottles (Table 3). The thickness of the mycoderm, however, is smaller than that of the latter, which is probably due to the fact that the mycoderm of *C. chanhua* cultivated with soil mulch adheres to, or is encased in, some small soil particles during the formation process, and is therefore of larger mass. However, the formation of *C. chanhua* mycoderm in soil mulching culture may be affected by soil factors. The growth of *C. chanhua* mycoderm in aseptic culture is unrestricted in glass bottles, so the thickness of soil mulched with mycoderm is significantly smaller than that of aseptic mycoderm.

### Hyphal arrangement and structure

As shown in Fig. 7, the mycoderm arrangement of *C. chanhua* mycoderm under aseptic cultivation and soil mulching cultivation is quite different under a scanning electron microscope. The hyphae of *C. chanhua* mycoderm under aseptic cultivation are arranged in a close and orderly manner (Fig. 7A–C), while the hyphae of *C. chanhua* mycoderm under soil mulching cultivation are arranged in a disorderly manner (Fig. 7G–I); this may be due to the complex soil environment, which has a certain blocking effect on the growth of hyphae, leading to the disordered arrangement of hyphae. However, the hyphae of *C. chanhua* under aseptic cultivation can grow smoothly without external force, so they are arranged in order. In addition, no matter what kind of environment, *C. chanhua* mycoderm is mostly connected by H-type hyphal bridge (Fig. 7D–F, J, and L).

### Hyphal density

It can be seen from Table 4 that the average number of hyphae of the soil-covered *C. chanhua* per slice is 17.33 and the density is 0.01289. The number of hyphae of *C. chanhua* cultivated in sterile glass bottle was 25.67, and the density was 0.01909. The analysis of variance showed that the density of hypha in sterile culture was significantly higher than that in soil mulching culture.

### Filtration effect of *C. chanhua* mycoderm on bacteria

Interestingly, we discovered that after the soil stock solution ($S_0$) is coated, the average total number of single colonies is 275.00 CFU/mL and the average total number of single colonies of 600 mesh nylon mesh filter solution ($S_1$) is 148.33 CFU/mL (Table 5), which is significantly different from the soil stock solution. This may be because some bacteria are filtered off by the nylon mesh or absorbed by the nylon mesh, resulting in a significant reduction in the number of bacteria. Nevertheless, after coating and culturing

**TABLE 3** *C. chanhua* mycelium cortices weight and thickness under different cultivation environments[a]

| Group | Sterile mycelium cortices | Soil-covered mycelium cortices |
|---|---|---|
| Fresh weight (g) | 0.4155 ± 0.09a | 0.5590 ± 0.08b |
| Dry weight (g) | 0.0978 ± 0.10a | 0.1563 ± 0.16b |
| Cortical thickness (mm) | 0.33–4.71 | 0.15–1.14 |

[a]Different lowercase letters represent statistically significant differences ($a = 0.05$).

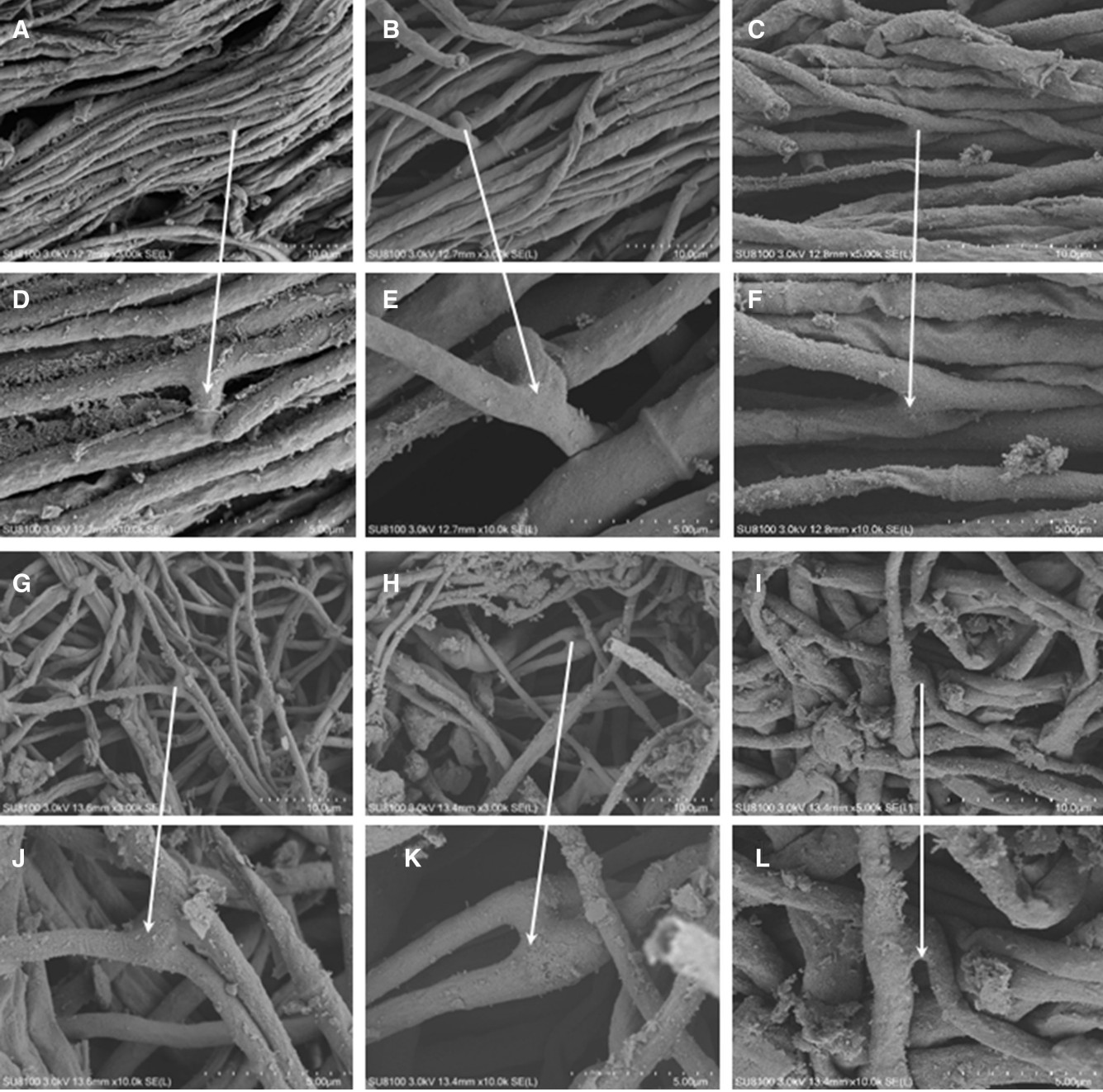

**FIG 7** Scanning electron microscope pictures of *C. chanhua* mycoderm. (A–C) 10 µm and (D–F) 5 µm: the arrangement and connection of the hyphae of *C. chanhua* in sterile glass bottle cultivation under scanning electron microscope. (G–I) 10 µm and (J and L) 5 µm: the arrangement and connection of the mycelium of the *C. chanhua* under the scanning electron microscope.

the mycoderm-filtered soil suspension ($S_2$), only a few single bacterial colonies grew, with an average value of 3.33 CFU/mL, indicating that most of the bacteria failed to pass through the mycoderm. There is no single bacterial colony growth in the sterile water filtrate ($S_{ck}$) filtered by mycoderm, eliminating the influence of bacteria in the mycoderm. It can also be seen from the growth of microorganisms in the culture medium that the number of bacteria is $S_0 > S_1 > S_2 > S_{ck}$. After coating and culturing the soil stock solution and sterile water filtrate filtered by the bacterial mycoderm, some *C. chanhua* spore powder germinates and grows in the culture medium (Fig. S5).

**TABLE 4** Hypha density of *C. chanhua* mycelium cortices under different cultivation environments[a]

| Name | Hyphal number | | | Average value | Length | Width | Dimension | Density |
|---|---|---|---|---|---|---|---|---|
| | 1 | 2 | 3 | | | | | |
| Soil-covered cultivated *C. chanhua* | 17 | 19 | 16 | 17.3333 | 31.77 | 42.32 | 1,344.506 | 0.01289 ± 0.001a |
| Glass bottle cultivated *C. chanhua* | 29 | 28 | 20 | 25.6667 | 31.77 | 42.32 | 1,344.506 | 0.01909 ± 0.003b |

[a]Use the 3.0k 10-μm scale as the standard to calculate the number of hyphae in the area of each slice. Different lowercase letters represent statistically significant differences ($a = 0.05$).

## The *C. chanhua* mycoderm can transport nitrogen

### Distribution and flow direction of $^{15}N$ in insects, mycoderms and soil

After inoculation of silkworm pupa with L-glutamine-$^{15}N$ + conidia suspension, the total nitrogen content in the body of the stiff worm is 10.16%, and the atomic percentage of $^{15}N$ is 0.3785% (Table S1). After mulching cultivation, the total nitrogen content in the mycoderm formed in the later stage was 8.12%, and $^{15}N$ was detected in both the mycoderm and the unlabeled soil, indicating that *C. chanhua* could transfer $^{15}N$ atoms out of the insect body with the formation of the hypha of the mycoderm. Although the $^{15}N$ content in the soil of the treatment group and the control group had no significant difference, the nitrogen content in the soil of the treatment group was higher than that of the control group; these results indicated that the presence of *C. chanhua* mycoderm promoted the transportation of nitrogen.

The $δ^{15}N$ values can be used to determine the enrichment degree of $^{15}N$ isotope in each sample compared with the standard, and the results show that $δ^{15}N$ is positive, indicating that $^{15}N$ is enriched in each sample to a certain extent. Before soil covering, sclerotia in *C. chanhua* $δ^{15}N‰$ value is 26.45‰; the $δ^{15}N$ values of inner sclerotia and mycoderm are 32.56‰ and 33.05‰, respectively. The concentration of $^{15}N$ in *C. chanhua* samples was significantly higher than that in soil (Fig. 8A), while there was no significant difference in the concentration of $^{15}N$ in soil between the experimental group and the control group, which may be due to the long-term cultivation of *C. chanhua* and the space limitation of plastic cups, resulting in a small number of microorganisms in the soil, which led to less absorption of nitrogen in the soil.

### Distribution and flow direction of soil $^{15}N$ in soil, mycoderm, and insect

The percentage of $^{15}N$ atoms in the soil treated with L-glutamine-$^{15}N$ is 0.4003% (Table S2). After the *C. chanhua* inoculated and injected stiff insects are covered with soil, the percentage of $^{15}N$ in the soil is significantly reduced to 0.3765%; the results showed that $^{15}N$ in the soil had obviously transferred and $^{15}N$ was detected in the mycoderm and inner sclerotia samples, proved that *C. chanhua* absorbed nitrogen in the soil during the growth process and the $^{15}N$ content in the mycoderm was significantly higher than that in the inner sclerotium samples, and showed that *C. chanhua* mainly absorbed nutrients from the surface mycoderm for the growth and reproduction of its own mycoderm.

The $δ^{15}N$ value was D1 > D2 > D3, which indicated that the enrichment of $^{15}N$ in soil, mycoderm, and inner sclerotia decreased significantly in turn (Fig. 8B). In the sclerotium sample (D3) in the treatment group, $δ^{15}N$ is only 0.58‰, indicating that $^{15}N$ is slightly

**TABLE 5** Number of bacterial colonies in a single plate treated in each experiment[a]

| Experimental treatment | Number of single colony of bacteria (CFU/mL) | | | Average value |
|---|---|---|---|---|
| | 1 | 2 | 3 | |
| S0 | 255 | 275 | 295 | 275.00 ± 20.00a |
| S1 | 175 | 150 | 120 | 148.33 ± 27.54b |
| S2 | 5 | 5 | 0 | 3.33 ± 2.89c |
| Sck | 0 | 0 | 0 | 0 |

[a]($S_0$) $10^{-7}$ concentration of soil liquid; ($S_1$) 600 mesh nylon mesh-filtered soil suspension; ($S_2$) soil suspension filtered by the fungus mycelium cortices of *C. chanhua*; and ($S_{ck}$) sterile water filtrate filtered by the mycelium cortices of *C. chanhua*. Different lowercase letters represent statistically significant differences ($a = 0.05$).

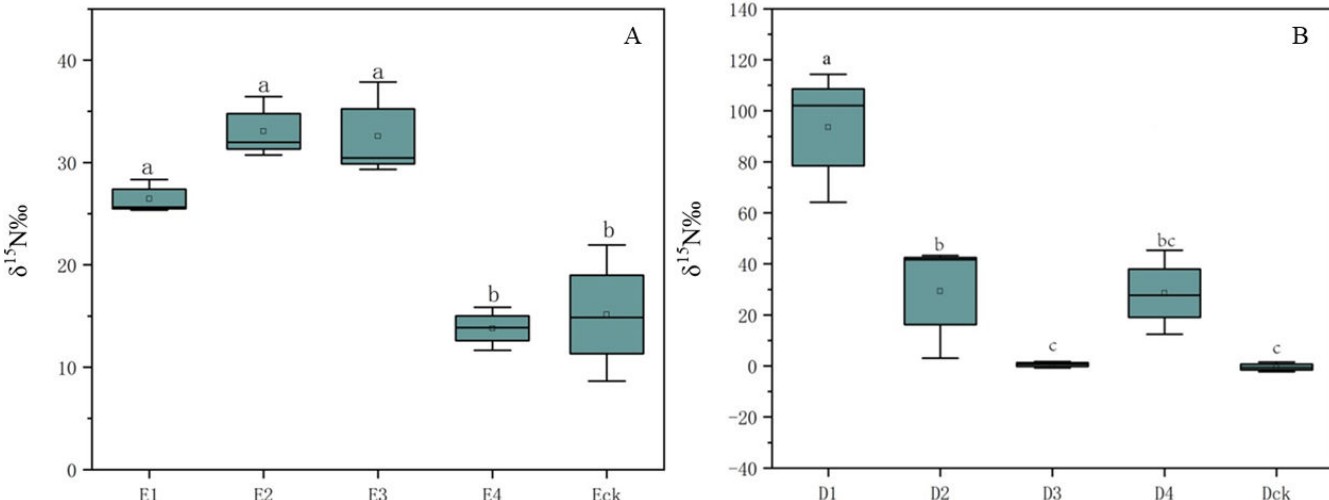

**FIG 8** $\delta^{15}N$ values in different samples in Experiments D and E. (A) $\delta^{15}N$ flows from *C. chanhua* to soil samples. (B) $\delta^{15}N$ flows from soil to *C. chanhua* samples. Note: (E1) L-glutamic acid-$^{15}$N + *C. chanhua* injected into silkworm pupa; (E2) *C. chanhua* membrane; (E3) internal sclerotium sample; (E4) soil within 2 cm of the *C. chanhua*; and (Eck) no injection of *C. chanhua* body soil within 2 cm. (D1) L-glutamic acid-$^{15}$N-treated labeled soil; (D2) *C. chanhua* membrane; (D3) internal sclerotium sample; (D4) soil within 2 cm of the *C. chanhua*; and (Dck) the body of the *C. chanhua* not injected. Different lowercase letters represent statistically significant differences ($a = 0.05$).

enriched, but in the control group, the $\delta^{15}N$ value is negative −0.41‰, indicating that the $^{15}$N isotope is depleted compared with the standard.

## DISCUSSION

Contact angle is a commonly used indicator to measure the degree of surface hydrophobicity, surface energy, heterogeneity, and roughness (31). Chau et al. (32) evaluated that the water contact angle of fungi was greater than 90°, and the fungi showed hydrophobicity; and the larger the contact angle, the stronger the hydrophobicity. The results of this study showed that the contact angle of *C. chanhua* under aseptic culture and soil-covered culture was greater than 90°, indicating the hydrophobicity of *C. chanhua* membrane, and it is the key to its survival and adaptation to the environment. The surface hydrophobicity of fungi enables them to better adsorb organic substances (33), which can be used as one of the surface characteristics affecting the adhesion and transportation of fungi in heterogeneous media (34). Fungi reproduce by producing hyphal networks and secrete enzymes to degrade complex nutrients into monosaccharides to maintain cell growth (35). The secretion of hydrophobin can significantly reduce the surface tension, enabling the mycelium to escape from the liquid and grow into the air (36). Fungal surface hydrophobic proteins are involved in various functions of fungal growth and development and help conidia adhere to insect epidermis (37), which has been reported in *Metarhizium anisopliae* and *Beauveria bassiana* (38, 39).

Through scanning electron microscope observation, it can be seen that the sterile-cultivated *C. chanhua* mycoderm is closely arranged and the soil-covered *C. chanhua* mycoderm is relatively sparse and disordered. This is because the soil-covered *C. chanhua* has certain resistance to growth in the soil, while the sterile-cultivated *C. chanhua* can grow in the air. Previous studies have observed that there were H-type hyphal fusions among the mycoderm hyphae of *C. sinensis* (40). Interestingly, we also observed that a large number of H-type hyphal fusions were also observed in the mycelia of *C. chanhua*. Therefore, there are a large number of H-type hyphal fusions in the mycelia of entomopathogenic fungi, and H-type hyphal fusions are often closely related to parasexual reproduction (41). Parasexual reproduction is widespread, which is a special reproductive mode occurring in fungi and an important source of fungal genetic variation (41, 42). Through parasexual reproduction, the genetic material between individuals can

communicate with each other, which is more convenient for the exchange of genetic material between individuals, which is easier for fungi to obtain and establish population advantages than sexual reproduction (43). In a complex environment, the formation of fungal mycoderm is affected by many factors, such as the roughness of the attachment surface, environmental factors, hydrophobicity, and other factors (44). In this study, scanning electron microscopy was used to observe the arrangement of the hyphae of *C. chanhua* and calculate their hyphal density. Our research results showed that the bacteria-membrane mycelia of aseptically cultivated *C. chanhua* were closely arranged, and the growth density of mycelia was larger than that of soil-covered *C. chanhua*. While the mycelia of soil-covered *C. chanhua* were relatively sparse and disorderly, the density was smaller, which was speculated to be related to the growth environment. There is a certain resistance to the growth of soil-covered *C. chanhua* in the soil, but the sterile-cultivated cicadas can grow smoothly in the air, so the mycelia are closely arranged. We can continue to explore the influence of different environmental factors on the formation of *C. chanhua* membrane in the future, which will help us understand the adaptation mechanism of *C. chanhua* to the environment and promote the ecological simulation of *C. chanhua*.

This study found that different cultivation environments had a greater impact on the composition and diversity of the bacterial community of *C. chanhua*; the finding indicated that external microbial factors had a greater impact on the bacterial community composition of entomopathogenic fungi. The common genera of all *C. chanhua* samples were analyzed, and the common bacteria included 22 genera, such as *Achromobacter*, *Serratia*, *Pseudomonas*, *Silesia*, *Enterococcus*, and *Bacillus*, indicating that these bacterial groups played an important role in the growth and development of *C. chanhua* and these bacterial groups were also found in wild *C. chanhua* samples (29). Previous studies have reported that the competition between microorganisms can promote the production of new active substances or increase the content of existing metabolites (45–47). For example, the production of HEA in *C. chanhua* was significantly increased after co-culturing bacteria of the genera *Cedecea*, *Serratia*, and *Enterococcus* with *C. chanhua* (48). In addition, studies have shown that *Serratia* not only has antibacterial activity but also has pathogenicity to some insect leaves (49, 50), so it is speculated that this kind of bacteria may participate in the process of *C. chanhua* infecting the host.

In this study, we found that with the growth of *C. chanhua*, more and more OTUs were found in the membrane of *C. chanhua*. It is reported that bacteria can move with the help of fungal hyphae (51, 52), so it is inferred that some bacteria in the mycoderm may come from the inner sclerotia. As the interface between *C. chanhua* and the environment, the mycoderm is a direct contact (29). We found that when the *C. chanhua* matured, the bacterial community on the mycelium was higher than the inner sclerotium. Therefore, it is speculated that the mycoderm may act as a barrier to prevent foreign bacteria from entering the inner sclerotia, thus playing a role in protecting *C. chanhua*. Based on it, we used *C. chanhua* mycoderm as a filter medium to filter soil bacteria and verified its biological filtering effect on bacteria. According to our results, only a small amount of bacteria grew in the soil suspension filtered by the mycoderm after the culture dish was coated, indicating that the mycoderm could filter most of the bacteria in the soil. A large number of environmental bacteria have been reported to colonize the mycoderm surface of *C. chanhua* (3). Although *C. chanhua* mycoderms grow in a complex soil environment, they can protect themselves from external pathogens, which is related to the characteristics of bacterial mycoderm (24). This mycorrhizal sheath structure with similar characteristics can enhance the disease resistance and stress resistance of host plants by absorbing and storing nutrients and effectively isolating pathogens (53). We found that *C. chanhua* mycoderm has a biological filtering effect on soil bacteria for the first time, which is helpful in understanding the relationship between microorganisms in the internal and external environment of *C. chanhua* and is of great significance for *C. chanhua* cultivation.

It is widely known that plants can obtain nitrogen in phytophagous insects through entomopathogenic fungi (54), and Behie et al. (55) found that *M. anisopliae*, *B. bassiana*, and *Lecanicillium lecanii* all showed the ability to transfer nitrogen to plants. In addition, *Metarhizium robertsii* has been reported to transfer nitrogen from the larvae of the wax borer to the plants of switchgrass for nitrogen fixation after infecting the larvae of the wax borer (56). It is interesting that the hyphae can transfer nitrogen from the insect body to the external environment and also can transfer nitrogen from the soil to the insect body. It has been reported that the entomopathogenic fungus *C. chanhua* is closely related to mycorrhizal fungi, which can transport nitrogen (28, 56). Previous studies have found that the mycoderm hyphae of entomopathogenic fungi can transfer nutrients from insects (57), and entomopathogenic fungi can also transfer nitrogen from infected insects to plants (58). Some entomopathogenic fungi, such as *M. anisopliae*, *B. bassiana*, and *Verticillium cerasus*, have the ability to transfer nitrogen (59). It has been reported that plants can obtain insect nitrogen sources by combining them with an endophytic entomopathogenic fungus (59). Behie and Bidochka (60) showed that all *M. anisopliae* and *B. bassiana* strains can transfer a large amount of insect-derived nitrogen to plants. *C. chanhua* is rich in nitrogen and during its growth and development, it will remove excess nitrogen from its body to ensure normal growth (61, 62).

In summary, this study is the first to verify that the membrane of *C. chanhua* has a filtration effect on bacteria and a transport function for nitrogen. In future studies, we will further explore which microorganisms *C. chanhua* can filter out and the specific forms that can transport nitrogen. This will help to understand the element network distribution mechanism of *C. chanhua* and provide a theoretical basis for the ecological cultivation technology of *C. chanhua*.

## Conclusion

Our study demonstrated for the first time to our knowledge that the *C. chanhua* mycoderm of soil-covered cultivation and sterile glass bottle cultivation was hydrophobic. In addition, *C. chanhua* mycoderm not only has the biological filtration function to soil bacteria, but also has the transport function to nitrogen element in the insect body and soil.

## MATERIALS AND METHODS

### Materials

#### Strain

*C. chanhua* strain (strain number.: GZUIFR_DJS1) is provided by the Institute of Fungal Resources, College of Life Sciences, Guizhou University.

#### Culture medium

Potato dextrose agar (PDA) medium is as follows: 200 g potato, 20 g glucose, 18 g agar powder, dilute to 1,000 mL with deionized water.

LB culture medium is as follows: 10 g tryptone, 5 g yeast extract, 10 g NaCl, 18 g agar powder, dilute to 1,000 mL with deionized water.

The above culture media were sterilized at 0.1 MPa for 30 min.

#### Main reagent

The E.Z.N.A.SoiL DNA Kit was provided by American OMEGA Company. The required L-Glutamine-15N was provided by Shanghai yuanye Bio-Technology Co., Ltd; the required anhydrous ethanol and isoamyl acetate was provided by Sinopharm Chemical Reagent Co., Ltd. The 15N stable isotope standard UREA4 (15N Atom% theoretical value: 0.366) was supplied by Elemental Trace Limited in the UK. Electron microscope fixative and PBS buffer were also used in this experiment and were provided by Xavier Biological

Company. Osmic acid was supplied by Ted Pella Inc. Ltd., and, $^{15}$N stable isotope standard UREA4 (theoretical value of $^{15}$N atom%: 0.366), was provided by Elemental Microanalysis Ltd., UK.

## Methods

### *C. chanhua cultivation*

*C. chanhua* strain was inoculated on PDA medium and activated in a constant temperature incubator at 25°C, and 0.05% Tween 80 solution was used to prepare the target concentration of $5 \times 10^7$ conidia/mL. Pupae of *Antheraea pernyi* Guérin-Méneville, sourced from the Dandong City Artificial Sericulture Base (Liaoning, China), were selected as host insects (cultivation medium) for *C. chanhua*, referring to Zeng et al. (3) method of surface disinfection of *A. pernyi*: wash the surface of *A. pernyi* with tap water, and dry it naturally; soak in 75% alcohol in the sterile super clean table, and scrub the insect surface with cotton for about 10 s; then rinse it with sterile water, clip it into a white porcelain plate covered with sterile paper towels, and dry the surface water; and suck 0.02 mL of prepared conidia suspension, and inject it into the third ring just behind the wing of the tussah pupa (63). The inoculated silkworm pupa is clamped into a sterile glass bottle and cultured in a dark place at 25°C. The bottle is moisturized with sterile wet cotton, and the cultivation process of *C. chanhua* is shown in Fig. S6.

### *Microbial diversity of C. chanhua*

*A. pernyi* pupa is infected by *C. chanhua* fungus after being cultivated in a glass bottle for 1 week to form a stiff worm (stiff worm stage). Take out the worm body, break off the inner sclerotium sample 5 g, and put it in a sterile centrifuge tube, with the label marked Si_1. Five silkworm pupae are mixed into one sample for three replications. After being ground evenly with liquid nitrogen, it is stored at −20°C. Some of the zombies will continue to be cultivated in sterile glass bottles, and the other part will be covered with soil under the masson pine forest of Guizhou University, with a thickness of 1–2 cm. After about 20 days of continuous cultivation in glass bottles, a layer of mycoderm formed by white mycelium (mycoderm formation stage) on the surface of *C. chanhua* was formed. The mycoderm was gently peeled off on the ultraclean workbench with sterile tweezers and put into a 10-mL centrifuge tube, which was recorded as Ci_2. Break off the insect body, and take the internal sclerotium sample in the centrifuge tube, which is recorded as Si_2. After 45 days of continuous cultivation in glass bottles, when the fruiting bodies of *C. chanhua* are almost mature (*C. chanhua* mature), cut off the fruiting bodies, take the samples of mycoderm (Ci_3) and inner sclerotia (Si_3) as before, and store them in centrifuge tubes at low temperature. Similarly, the samples of bacterial mycoderm (Cs_2) and inner sclerotia (Ss_2) of *C. chanhua* cultivated with soil mulching were taken at the same time as those of *C. chanhua* in aseptic glass bottles during the formation of bacterial mycoderm, and the samples of bacterial mycoderm (Cs_3) and inner sclerotia (Ss_3) were collected at the mature stage for cryopreservation. All samples were collected and ground evenly with liquid nitrogen and then stored at −20°C until DNA was extracted.

### *The morphology of the mycelium of C. chanhua in the soil and atmospheric environment*

After *C. chanhua* is mature, collect the fresh *C. chanhua* cultivated in soil covering and sterile glass bottle respectively, and take it back to the laboratory. After washing the surface soil, use sterile paper to absorb the surface water, cut off the fruiting body, and gently peel off the bacterial mycoderm with tweezers. First, mix every five *C. chanhua* mycoderms into a group, repeat three times, weigh with a precision balance, record the fresh weight, and then weigh and record the dry weight after drying at 60°C. Then, observe the thickness of *C. chanhua* mycoderm under stereomicroscope, and use a scanning electron microscope to observe the structure, density, and arrangement of

*C. chanhua* mycoderm under soil and air environment according to the method of Jackowiak, and see Table S3 for specific steps (64). Finally, the contact angle measuring device was used to determine the hydrophilicity/hydrophobicity of the *C. chanhua* mycoderm by using the method of Huhtamäki et al. (65).

### Filtration effect of C. chanhua mycoderm on bacteria

Soil was randomly dug from the ground of masson pine forests in Guizhou University (Guiyang, Guizhou) and taken back to the laboratory; 10 g of fresh soil was weighed and added into 90 mL of sterile water. It was incubated at 25°C in a shake flask for 2–3 h. It was filtered into a new sterile triangular flask with sterile gauze. The soil suspension was diluted to $10^{-7}$ in turn for standby.

   Three groups of different treatments are set up in this experiment.

1. Group A: 600 mesh sterile nylon mesh cut into $7 \times 7$ cm$^2$ is folded after sterilization and put into a sterile funnel. One milliliter of soil suspension is sucked each time for filtration. The filtrate is collected with a sterile centrifuge tube until 5 mL of filtrate is collected.

2. Group B: spread the *C. chanhua* mycoderm peeled with sterile tweezers to about 3 × 3 cm, put it into the sterile funnel, so that the mycoderm forms a concave surface at the funnel tube, then use the pipette gun to suck the soil filtrate, and slowly drop it into the concave surface of the mycoderm (the soil suspension does not exceed the concave surface), so that it can be filtered slowly. Collect about 5 mL of filtrate with a sterile centrifuge tube.

3. Control group (CK group): in the same way as Group B, sterile water was filtered by mycoderm, and 5 mL of filtrate was collected by centrifuge tube for standby.

   Soil suspension stock (S0) with 10-7 concentration, nylon mesh filtrate (S1), membrane filtrate (S2) and membrane filtered sterile water (Sck) were absorbed and coated on the medium with a 200 µL pipette, and the bacteria were cultured at 25°C, with 3 repeats for each treatment. After 2 days of incubation, the single bacterial colony in each culture dish was counted. The experimental process of bacterial filtration by *C. chanhua* mycoderm is shown in Fig. S7.

### $^{15}$N flows from the soil to the insect body through the C. chanhua fungus mycoderm (recorded as Experiment D)

Add 10 kg fresh soil dug from the masson pine forest to each cultivation basin, balancing the humidity for 3 days, so that its physical and chemical properties are consistent. Use sterile water to prepare 100 mg/L L-glutamine-$^{15}$N solution. After it is completely dissolved, shake it up. Add 500 mL of the prepared L-glutamine-$^{15}$N solution to each pot of soil. Mix the soil every day for 3 weeks. During the balance process, use a weighing method to supplement soil water. After the L-glutamine-$^{15}$N is supplied, take the soil sample, and store it at −20°C, which is recorded as D1.

   Add L-glutamine-$^{15}$N-treated soil into the plastic cup (the soil volume is about 80%), and press it as tightly as possible. The hole punch will punch a hole with a diameter of 2.5 cm in the middle. The rigors inoculated with a spore suspension concentration of 5 × $10^7$ conidia/mL were placed into soil holes, the surface was covered with 1- to 2-cm-thick soil, the surface of the plastic cup was covered with a preservative film, the hole was tied with a toothpick, and it was cultivated in a dark place at 25°C for 1 month. Rinse the surface soil of mature *C. chanhua*, and then suck up the surface water with sterile paper. After cutting off the fruiting bodies, tear off the mycoderm, and collect the samples of *C. chanhua* mycoderm (D2) and inner sclerotia (D3) on the ultrapure clean workbench in a sterile centrifuge tube. Scrape the soil (D4) samples within 2 cm around *C. chanhua* into a ziplock bag. Figure S8 shows the experimental process and sampling diagram of $^{15}$N flowing from soil to insect body through *C. chanhua* mycoderm. Five treatments were

mixed into one sample, three of which are repeated, and stored at −20℃. After the live tussah pupa is disinfected on the surface, it is covered with soil treated with isotopes. After the same treatment time as the experimental group, the tussah pupa is taken as the control group sample (Dck), and the $^{15}$N abundance value is compared with the treatment group.

### $^{15}$N flows from the insect body to the soil through the C. chanhua mycoderm (recorded as Experiment E)

Conidial suspension with a concentration of $5 \times 10^7$ conidia/mL was prepared, and 8 mg of L-glutamine-$^{15}$N was added to each milliliter of spore suspension, which was completely dissolved and shaken well. Inject 0.02 mL conidia suspension + L-glutamine-$^{15}$N solution into the third ring just behind the wing of each silkworm pupa, clip it into a sterile glass bottle with sterile tweezers, and keep it moist at 25℃ for 5–7 days. After the silkworm pupa is infected and forms a stiff worm, take out the inner sclerotium sample for preservation, and record it as E1.

Put fresh soil into a disposable plastic cup with a soil volume of about 80%, and try to compress the soil. Punch a hole with a diameter of 2.5 cm in the middle of the plastic cup containing soil. Put the L-glutamine-$^{15}$N-labeled zombie into the soil hole, and cover the surface with about 2 cm of soil. Cover the surface of the plastic cup with a fresh-keeping film, and tie a hole to keep moisture and breathable. Cultivate at 25℃ in a dark place for 1 month. After *C. chanhua* matures, collect *C. chanhua* mycoderm (E2), inner sclerotium (E3), and soil (E4) samples within 2 cm around *C. chanhua*, and store them at −20℃.

Add 8 mg L-glutamine-$^{15}$N to each 1 mL of sterile water, shake well after it is completely dissolved, inject 0.02 mL of this solution into each silkworm chrysalis, and cover it with soil at the same time as the experimental group. After the same treatment time as the experimental group, take soil samples (Eck) within 2 cm around the insect body to verify the transportation of $^{15}$N isotopes of the insect body to the soil in the presence of *C. chanhua* mycoderm. The experimental process and sampling diagram of the flow of $^{15}$N from the insect body to the soil through the *C. chanhua* mycoderm are shown in Fig. S9.

The test samples were sent to Jiangsu Nanjing Carvensys Testing Technology Co., Ltd. for determination of $^{15}$N content, with detection equipment SerCon Integra 2 Integrated EA-IRMS.

### Total bacterial DNA extraction, PCR amplification, and high-throughput sequencing

E.Z.N.A. Soil DNA Kit is used to extract the total microbial DNA according to the operation procedure. The upstream and downstream primers 338F (5′-ACTCCTACGGGGAGGCAG CAG-3′) and 806R (5′-GACTACHVGGGTWTCTAA T-3′) were used for PCR amplification. Use the PCR instrument (ABI Gene Amp 9700) for PCR reaction parameters: 95℃ for 3 min, 95℃ for 30 s, 55℃ for 30 s, 72℃ for 45 s, and 30 cycles. At 72℃, the reaction was prolonged for 10 min until the reaction stopped at 10℃ (66). The amplified products were sent to Illumina MiSeq sequencing platform of Shanghai Meiji Biomedical Technology Co., Ltd. for high-throughput sequencing.

### Sequence processing and OTU comments

The PE reads obtained by MiSeq sequencing are spliced according to the overlap relationship, and the sequence quality is controlled and filtered at the same time. Then, according to the barcodes and primer sequences at both ends of the sequence, we can distinguish samples to get effective sequences and correct the sequence direction to get the optimized sequence. After all the original sequences were filtered and optimized and the chimeras were removed, OTU clustering was conducted on the QIIME platform and compared with the UNITE fungal database to obtain the taxonomic information of each OTU and make taxonomic analysis based on representative sequences (67).

## Diversity analysis

All samples shall be homogenized according to the minimum sequence number. OTU with 97% similarity shall be selected, and Mothur method shall be used (http://www.mothur.org/wiki/). Calculate the Ace index, Shannon index, Simpson index, Chao index, and Coverage index of each group. In this study, the sclerotium and mycoderm samples in the stage of sclerotia, sterile cultivation, and mulching cultivation in the stage of mycoderm formation and maturity were divided into nine groups of samples, with three replicates in each group. The average diversity index of each group was calculated by SPSS software, and the significant differences between the groups were compared by one-way analysis of variance (ANOVA). Using the unweighted group average method to cluster and calculate the beta diversity index of different samples, principal coordinate analysis (PCoA) was conducted to clarify the distance matrix between different groups.

## Analysis of bacterial community structure

Use the R language tool to make a percentage stacking column chart for the bacterial groups whose relative abundance is greater than 1.0% in each group of samples. If the relative abundance is less than 1.0%, it should be attributed to others (the relative abundance is the average of three repeats of each group of samples). The common OTU and unique OTU of *C. chanhua* samples in different stages under the same environment and in different cultivation environments at the same stage were analyzed by Venn diagram, and the similarity and overlap of species composition of each sample were analyzed. The petal map was drawn for all *C. chanhua* samples, and the common and endemic genera in each sample were analyzed; based on the bacterial community abundance data in the samples, the significant differences between groups of the top 15 genera of bacterial community abundance in different samples were analyzed using one-way ANOVA, and significant species of differences between sample groups were obtained.

## Bacterial function prediction

Via PICRUSt (http://picrust.github.io/picrust/), the 16S series of bacteria were annotated with Clusters of Orthologous Groups of proteins (COG) and Kyoto Encyclopedia of Genes and Genomes (KEGG) functions to obtain the annotation information of OTU at each functional level of COG and KEGG and its abundance in different samples.

## Data analysis

Other test data were collated by Excel and statistically analyzed by SPSS software.

### ACKNOWLEDGMENTS

This work was supported by grants from the National Natural Science Foundation of China (no. 31860037), Department of Science and Technology of Guizhou Province (no. [2020]1Z009), and Guizhou Qianxinan Tobacco Company (no. 2022-04).

### AUTHOR AFFILIATIONS

[1]Institute of Fungal Resources, College of Life Sciences, Guizhou University, Guiyang, Guizhou, China
[2]Department of Humanities, Business College of Guizhou University of Finance and Economics, Qiannan, Guizhou, China
[3]Tea College, Guizhou University, Guiyang, Guizhou, China

### AUTHOR ORCIDs

Gongping Hu  http://orcid.org/0009-0000-9356-2683
Xiao Zou  http://orcid.org/0000-0002-3666-5536

## FUNDING

| Funder | Grant(s) | Author(s) |
|---|---|---|
| Foundation for Innovative Research Groups of the National Natural Science Foundation of China | 31860037 | Xiao Zou |
| 贵州省科技厅 \| Science and Technology Program of Guizhou Province (贵州省科技计划项目) | [2020]1Z009 | Xiao Zou |
| Guizhou Tobacco Company Qianxinan Branch | 2022-04 | Xiao Zou |

## AUTHOR CONTRIBUTIONS

Gongping Hu, Writing – original draft.

## DATA AVAILABILITY

The raw sequence data have been deposited in the SRA database under accession number PRJNA944443.

## ADDITIONAL FILES

The following material is available online.

### Supplemental Material

**Supplemental material (Spectrum01179-23-s0001.docx).** Fig. S1 to Fig. S9; Tables S1 to S3.

### Open Peer Review

**PEER REVIEW HISTORY (review-history.pdf).** An accounting of the reviewer comments and feedback.

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
