## [Reviewer comments · Microbiology Spectrum]

Microbiology Spectrum

Filtration Effect of *Cordyceps chanhua* Mycoderm on Bacteria and its Transport Function on Nitrogen

Gongping Hu, Yeming Zhou, Dan Mou, Jiaojiao Qu, Li Luo, Lin Duan, Zhongshun Xu, and Xiao Zou

Corresponding Author(s): Xiao Zou, Guizhou University

Review Timeline:

Submission Date:	March 19, 2023
Editorial Decision:	April 22, 2023
Revision Received:	September 21, 2023
Editorial Decision:	October 9, 2023
Revision Received:	October 10, 2023
Editorial Decision:	October 14, 2023
Revision Received:	October 17, 2023
Editorial Decision:	October 22, 2023
Revision Received:	October 30, 2023
Accepted:	November 4, 2023

Editor: Chengshu Wang

Reviewer(s): Disclosure of reviewer identity is with reference to reviewer comments included in decision letter(s). The following individuals involved in review of your submission have agreed to reveal their identity: Caihong Dong (Reviewer #1); Bo Huang (Reviewer #2)

Transaction Report:

DOI: <https://doi.org/10.1128/spectrum.01179-23>

April 22, 2023

Dr. xiao Zou
Institute of Fungus Resources
Guizhou University
guiyang
China

Re: Spectrum01179-23 (Filtration Effect of *Cordyceps cicadae* Mycoderm on Bacteria and its Transport Function on Nitrogen)

Dear Dr. xiao Zou:

Link Not Available

Sincerely,

Chengshu Wang

Journals Department
Reviewer comments:

Reviewer #1 (Comments for the Author):

The article aimed to explore the role of *C. cicadae* mycoderm. It was found that the bacterial membrane of *C. cicadae* not only filters environmental bacteria, but also absorbs and transports nitrogen elements inside and outside the body of *C. cicadae*. This article presents some interesting data. However, it's not easy to understand the experimental and methodological methods. Maybe some figures or schematic diagrams should be added to make it clear, such as the filtration effect and N15 flow experiments. And also, the text is not set out well.

1. The Microbial Diversity is the diversity of mycoderm? Please specified throughout the text.
2. Why there is no results of fungal diversity.
3. If the Pupae of *Antheraea pernyi* is sterilized thoroughly before being infection, *C. cicadae* can develop normally? It should be

discussed.

4. *Cordyceps cicadae* should be *Cordyceps chanhua*.

L84 Bacterial Microbial Diversity of ??

L233 The formation of "H" type fusion hypha is conducive to the material exchange and signal transduction between their hypha. reference?

L255 Transport Function of *C. cicadae* Mycoderm to Nitrogen? What means?

L497 at the concentration of 500 mL??

Reviewer #2 (Comments for the Author):

In this manuscript, the authors analyzed the bacterial community in different parts of *C. cicadae* in different production stages and different environments by high-throughput sequencing. They also demonstrated that the filtration effect of *C. cicadae* membrane on bacterial communities. Furthermore, they also found the nitrogen transport function of the membrane of *C. cicadae*. The obtained information provide a basis for understanding the mechanism of the internal microbial community maintained by *C. cicadae* and nutritional balance by self-regulation in *C. cicadae*.

Specific comments and suggestions:

1.

L68-71, it is confused for the researches on the morphological characteristics of the mycoderm in the sentence "most studies on the morphological characteristics of cordyceps fungi focus on the mycoderm, sclerotia, stroma and ascus structures of cordyceps (24, 25, 26). However, as a special structure of some entomopathogenic fungi, the study on the morphological characteristics of the mycoderm has not been reported." The authors should explain and revise.

2.

L206, the subtitle for "Apparent Morphology of *C. cicadae* Mycoderm in Soil Atmosphere", it should be "Soil and Atmosphere".

3.

L240-248, It is better to provide the morphology features of CFU (or at least several representative images for the morphology character of CFU).

4.

L497, "at the concentration of 500 mL", it is not concentration.

5.

L538-539, The details for equipment and parameter should be provided.

6.

L863-866

What did the authors mean for the indicated arrow, the author should indicate.

7.

L881, the description should be used by English.

Staff Comments:

Preparing Revision Guidelines

- Point-by-point responses to the issues raised by the reviewers in a file named "Response to Reviewers," NOT IN YOUR COVER LETTER.
- Upload a compare copy of the manuscript (without figures) as a "Marked-Up Manuscript" file.
- Each figure must be uploaded as a separate file, and any multipanel figures must be assembled into one file.

- Manuscript: A .DOC version of the revised manuscript
- Figures: Editable, high-resolution, individual figure files are required at revision, TIFF or EPS files are preferred

Please return the manuscript within 60 days; if you cannot complete the modification within this time period, please contact me. If you do not wish to modify the manuscript and prefer to submit it to another journal, please notify me of your decision immediately so that the manuscript may be formally withdrawn from consideration by Microbiology Spectrum.

Response to Reviewers

We sincerely thank the editor and all reviewers for their valuable feedback that we have used to improve the quality of our manuscript. The reviewer comments are laid out below in italicized font and specific concerns have been numbered.

--The manuscript is missing legends for figures 10-13. Please provide figure legends for all figures in the manuscript.

We apologize for the lack of legends in the manuscript, we have now placed the legends and table before the references in the manuscript and marked them in red font.

October 9, 2023

Dr. Xiao Zou
Guizhou University
College of Life Sciences
Guiyang
China

Re: Spectrum01179-23R1 (Filtration Effect of *Cordyceps chanhua* Mycoderm on Bacteria and its Transport Function on Nitrogen)

Dear Dr. Xiao Zou:

Please follow the requirements and Reviewers' suggestions to make corrections and improvements. Multiple rounds of modifications are not permitted.

Link Not Available

Sincerely,

Chengshu Wang

Journals Department
Reviewer comments:

Reviewer #1 (Comments for the Author):

The file "response to reviewers" just show the response to editor and no response to reviewers. There is also no modifacaiton in the marked version except the figure legend.

Reviewer #2 (Comments for the Author):

1. The authors should provide the responses in point-by-point.
2. The authors should provide the changes with highlighting in the revised manuscript, which can clarify the review's concerns.
3. It should be "discussion" (Line 295).

Staff Comments:

Preparing Revision Guidelines

Please return the manuscript within 60 days; if you cannot complete the modification within this time period, please contact me. If you do not wish to modify the manuscript and prefer to submit it to another journal, please notify me of your decision immediately so that the manuscript may be formally withdrawn from consideration by Microbiology Spectrum.

Response to Reviewers

We sincerely thank the editor and all reviewers for their valuable feedback that we have used to improve the quality of our manuscript. The reviewer comments are laid out below in italicized font and specific concerns have been numbered. Our response is given in normal font and changes/additions to the manuscript are given in the red text.

Reviewer #1 (Comments for the Author)

It's not easy to understand the experimental and methodological methods. Maybe some figures or schematic diagrams should be added to make it clear, such as the filtration effect and N15 flow experiments. And also, the text is not set out well.

Response: We feel great thanks for your professional review work on our article. As you are concerned, there are several problems that need to be addressed. According to your nice suggestions, we have made corrections to our previous draft, the detailed corrections are listed below.

(1) We have corrected some textual errors in the article, and the changes to the manuscript have been shown in red text.

(2) In order to better understand the experiment and methodological methods, we drew some schematic diagrams in the manuscript. We have put schematic diagrams in the corresponding position in the manuscript.

1. The Microbial Diversity is the diversity of mycoderm? Please specified throughout the text.

We apologize for not specifying the meaning of microbial diversity in the text, and the microbial diversity in this study refers to the bacterial community diversity of internal sclerotium and mycoderm of *Cordyceps Chanhua* at different growth stages and under different cultivation environments, we have specified in the manuscript.

2. Why there is no results of fungal diversity.

In manuscript L71~73, previous studies have found that the relative abundance of other fungi in the sclerotia of *C. chanhua* was relatively low, while the bacterial

community was relatively rich, therefore, we didn't study fungal diversity.

3. If the pupae of *Antheraea pernyi* is sterilized thoroughly before being infection, *C. cicadae* can develop normally? It should be discussed.

Thank you very much for your valuable advice on our manuscript, to better answer this question, we designed an experiment on the effect of complete disinfection of the pupae of *Antheraea pernyi* on the growth and development of *C. chanhua*. We designed a total of 4 groups of experiments, namely (1) disinfection was carried out according to the disinfection method of the pupae of *Antheraea pernyi* in the original manuscript (d₁): wash the surface of *A. pernyi* with tap water and dry it naturally. Soak in 75% alcohol in the sterile super clean table and scrub the insect surface with cotton for about ten seconds, then rinse it with sterile water, clip it into a white porcelain plate covered with sterile paper towels, and dry the surface water; (2) the sodium hypochlorite solution was added on the basis of the experimental group d₁ (d₂): First rinse the surface of the surface of *A. pernyi* with tap water, dry naturally; the insect body surface was soaked with 75% alcohol and scrubbed with cotton for about 10 s in a sterile ultra-clean table, and then washed with 5% sodium hypochlorite solution for 10 s. The most sterile water was used for washing, and then placed on a white porcelain plate covered with sterile paper towels to absorb the surface moisture; (3) a slight improvement was made on the basis of experimental group d₂ (d₃): First rinse the surface of the surface of *A. pernyi* with tap water, dry naturally; the insect body surface was soaked with 75% alcohol and scrubbed with cotton for about 20 s in a sterile ultra-clean table, and then washed with 5% sodium hypochlorite solution for 20 s. The most sterile water was used for washing, and then placed on a white porcelain plate covered with sterile paper towels to absorb the surface moisture; (4) the surface of *A. pernyi* was completely sterilized at 121 °C for 30 min (d₄). The growth and development of *C. Chanhua* 25 days after inoculation were shown in Table 1.

Table 1 Growth information of *Cordyceps Chanhua* under different disinfection methods of the pupae of *Antheraea pernyi*

Method of disinfection	Total pupae	Number of fossilized pupae	Rate of rigidity	Number of fruiting bodies
d ₁	12	11	91.6%	8.66667±1.08012
d ₂	12	12	100.00%	8.88889±0.69611
d ₃	12	11	91.67%	8.77778±0.77778
d ₄	12	3	25.00%	5.00000±0.57735

As can be seen from Table 1, 25 days after inoculation, d₂ disinfection treatment had the highest rate of *C. Chanhua* rigidor and the highest number of fruite bodies, however, d₄ had the lowest ossification rate and number of frutes. Therefore, the pupae of *A. pernyi* sterilization is not suitable for *C. Chanhua* cultivation.

4. *Cordyceps cicadae* should be *Cordyceps chanhua*.

As you are concerned, *Cordyceps cicadae* should be *Cordyceps chanhua*. The reason is that Chinese scholar Li Zengzhi found the sexual *Cordyceps chanhua*, classified it as a new species of the genus *Cordyceps*, and named it *Cordyceps chanhua* with the ancient name of *Cordyceps cicadae* (1).

REFERENCE

1. Li ZZ, Luan FG, Nigel HJ, Zhang SL, Chen MJ, Huang B, Shun CS, Chen ZA, Li CR, Tan YJ, Dong JF. Biodiversity of cordycipitoid fungi associated with *Isaria cicadae* Miquel II: Teleomorph discovery and nomenclature of chanhua, an important medicinal fungus in China. *Mycosystema*, 2021, 40(2): 1-12. <https://doi.org/10.13346/j.mycosystema.200119>.

Thank you again for your careful checks. Based on your comments, we have corrected some other minor errors in the manuscript.

Reviewer #2 (Comments for the Author):

In this manuscript, the authors analyzed the bacterial community in different parts of C. cicadae in different production stages and different environments by high-throughput sequencing. They also demonstrated that the filtration effect of C. cicadae membrane on bacterial communities. Furthermore, they also found the

nitrogen transport function of the membrane of C.cicadae. The obtained information provide a basis for understanding the mechanism of the internal microbial community maintained by C. cicadae and nutritional balance by self-regulation in C. cicadae.

Specific comments and suggestions:

1. L68-71, it is confused for the researches on the morphological characteristics of the mycoderm in the sentence "most studies on the morphological characteristics of cordyceps fungi focus on the mycoderm, sclerotia, stroma and ascus structures of cordyceps (24, 25, 26). However, as a special structure of some entomopathogenic fungi, the study on the morphological characteristics of the mycoderm has not been reported." The authors should explain and revise.

Response: Thanks for your careful checks. We are sorry for our carelessness. Based on your comments, we have made the corrections to make the word harmonized within the whole manuscript.

2. L206, the subtitle for "Apparent Morphology of C. cicadae Mycoderm in Soil Atmosphere", it should be "Soil and Atmosphere".

Response: We have made the corresponding changes in the manuscript.

3. L240-248, It is better to provide the morphology features of CFU (or at least several representative images for the morphology character of CFU).

Response: We have included pictures of CFU morphological characteristics of soil bacteria cultured on plate medium in the supplementary material.

4. L497, "at the concentration of 500 mL", it is not concentration.

Response: Thanks for your careful checks. Based on your comments, we have made the corresponding changes in the manuscript.

5. L538-539, The details for equipment and parameter should be provided.

Response: We appreciate your valuable comments. We have added the specific equipment parameters of L538-539 to the corresponding positions in the manuscript, marked with red font

6. L863-866 What did the authors mean for the indicated arrow, the author should indicate.

Response: The arrows in the picture represent the "H" type fusion hyphae partial zoom. We have highlighted it in the notes.

7. L881, the description should be used by English.

Response: Thanks for your careful checks. We have made the corresponding changes in the manuscript.

8. It should be "discussion" (Line 295).

Response: Thanks for your careful checks. We have made the corresponding changes in the manuscript.

October 14, 2023

Dr. Xiao Zou
Guizhou University
College of Life Sciences
Guiyang
China

Re: Spectrum01179-23R2 (Filtration Effect of *Cordyceps chanhua* Mycoderm on Bacteria and its Transport Function on Nitrogen)

Dear Dr. Xiao Zou:

Thanks for this revision. Many improvements are still required before further suggestion, especially regarding the Table and Figure numbers:

1. *Cordyceps cicadae* is still used in Supplementary.
2. Merge Fig. 8 and Fig. 9 into a figure with two panels.
3. Move the Figures 10-13 mentioned in Methods to Supplementary and re-order them.

It is not appropriate to have quite a few (small) Tables, Table 8 should go to supplementary. What are the relationships regarding Table 6 vs. Fig. 7, and Table 7 vs. Fig. 9, the same things? If being similar, please move Tables 6 and 7 to supplementary as well.

Link Not Available

Sincerely,

Chengshu Wang

Journals Department
Reviewer comments:

Staff Comments:

Preparing Revision Guidelines

Please return the manuscript within 60 days; if you cannot complete the modification within this time period, please contact me. If you do not wish to modify the manuscript and prefer to submit it to another journal, please notify me of your decision immediately so that the manuscript may be formally withdrawn from consideration by Microbiology Spectrum.

Response to Reviewers

We sincerely thank the editor and all reviewers for their valuable feedback that we have used to improve the quality of our manuscript. The reviewer comments are laid out below in italicized font and specific concerns have been numbered. Our response is given in normal font and changes/additions to the manuscript are given in the red text.

Response to the reviewer on October 14

1. *Cordyceps cicadae* is still used in Supplementary.

Response: Thank you for your careful checks. And we've already changed *Cordyceps cicadae* to *Cordyceps chanhua* in the Supplementary.

2. Merge Fig. 8 and Fig. 9 into a figure with two panels.

Response: We have merged Fig. 8 and Fig. 9 into a figure with two panels, and the new number is Fig. 8 in the manuscript. The corresponding figures and legends have been placed after the references in the manuscript. The final result is shown below.

Fig. 8 $\delta^{15}\text{N}$ values in different samples in experiment D and experiment E. A: $\delta^{15}\text{N}$ flows from *Cordyceps chanhua* to soil samples. B: $\delta^{15}\text{N}$ flows from soil to *Cordyceps chanhua* samples.

3. Move the Figures 10-13 mentioned in Methods to Supplementary and reorder them.

Response: Thanks for your advice and we have moved the Figures 10-13 mentioned in Methods to Supplementary and re-order them. And Figures 10-13 Corresponding numbers in supplementary are shown in Fig. S6-S9.

4. It is not appropriate to have quite a few (small) Tables, Table 8 should go to supplementary. What are the relationships regarding Table 6 vs. Fig. 7, and Table 7 vs. Fig. 9, the same things? If being similar, please move Tables 6 and 7 to supplementary as well.

Response: Thanks for your careful checks. We have moved Table 8 into the supplementary, and the corresponding number in the supplementary material is Table S3. In addition, Table 6 and Table 7 show the ^{15}N and total nitrogen content of different samples in experiment D and experiment E, respectively. Fig. 8 and Fig. 9 (Now merged into Fig. 8) respectively show the enrichment degree of each sample of ^{15}N in experiment D and experiment E. We have now moved Table 6 and Table 7 into the supplementary, and the corresponding number in the supplementary material is Table S1 and Table S2.

Reviewer #1 (Comments for the Author)

It's not easy to understand the experimental and methodological methods. Maybe some figures or schematic diagrams should be added to make it clear, such as the filtration effect and N15 flow experiments. And also, the text is not set out well.

Response: We feel great thanks for your professional review work on our article. As you are concerned, there are several problems that need to be addressed. According to your nice suggestions, we have made corrections to our previous draft, the detailed corrections are listed below.

(1) We have corrected some textual errors in the article, and the changes to the manuscript have been shown in red text.

(2) In order to better understand the experiment and methodological methods, we drew some schematic diagrams in the manuscript. We have put schematic diagrams in the corresponding position in the manuscript.

Response to the reviewer on October 10

1. The Microbial Diversity is the diversity of mycoderm? Please specified throughout the text.

We apologize for not specifying the meaning of microbial diversity in the text, and the microbial diversity in this study refers to the bacterial community diversity of internal sclerotium and mycoderm of *Cordyceps Chanhua* at different growth stages and under different cultivation environments, we have specified in the manuscript.

2. Why there is no results of fungal diversity.

In manuscript L71~73, previous studies have found that the relative abundance of other fungi in the sclerotia of *C. chanhua* was relatively low, while the bacterial community was relatively rich, therefore, we didn't study fungal diversity.

3. If the pupae of *Antheraea pernyi* is sterilized thoroughly before being infection, *C. cicadae* can develop normally? It should be discussed.

Thank you very much for your valuable advice on our manuscript, to better answer this question, we designed an experiment on the effect of complete disinfection of the pupae of *Antheraea pernyi* on the growth and development of *C. chanhua*. We designed a total of 4 groups of experiments, namely (1) disinfection was carried out according to the disinfection method of the pupae of *Antheraea pernyi* in the original manuscript (d₁): wash the surface of *A. pernyi* with tap water and dry it naturally. Soak in 75% alcohol in the sterile super clean table and scrub the insect surface with cotton for about ten seconds, then rinse it with sterile water, clip it into a white porcelain plate covered with sterile paper towels, and dry the surface water; (2) the sodium hypochlorite solution was added on the basis of the experimental group d₁

(d₂): First rinse the surface of the surface of *A. pernyi* with tap water, dry naturally; the insect body surface was soaked with 75% alcohol and scrubbed with cotton for about 10 s in a sterile ultra-clean table, and then washed with 5% sodium hypochlorite solution for 10 s. The most sterile water was used for washing, and then placed on a white porcelain plate covered with sterile paper towels to absorb the surface moisture; (3) a slight improvement was made on the basis of experimental group d₂ (d₃): First rinse the surface of the surface of *A. pernyi* with tap water, dry naturally; the insect body surface was soaked with 75% alcohol and scrubbed with cotton for about 20 s in a sterile ultra-clean table, and then washed with 5% sodium hypochlorite solution for 20 s. The most sterile water was used for washing, and then placed on a white porcelain plate covered with sterile paper towels to absorb the surface moisture; (4) the surface of *A. pernyi* was completely sterilized at 121 °C for 30 min (d₄). The growth and development of *C. Chanhua* 25 days after inoculation were shown in Table 1.

Table 1 Growth information of *Cordyceps Chanhua* under different disinfection methods of the pupae of *Antheraea pernyi*

Method of disinfection	Total pupae	Number of fossilized pupae	Rate of rigidity	Number of fruiting bodies
d ₁	12	11	91.6%	8.66667±1.08012
d ₂	12	12	100.00%	8.88889±0.69611
d ₃	12	11	91.67%	8.77778±0.77778
d ₄	12	3	25.00%	5.00000±0.57735

As can be seen from Table 1, 25 days after inoculation, d₂ disinfection treatment had the highest rate of *C. Chanhua* rigidor and the highest number of fruite bodies, however, d₄ had the lowest ossification rate and number of fruites. Therefore, the pupae of *A. pernyi* sterilization is not suitable for *C. Chanhua* cultivation.

4. *Cordyceps cicadae* should be *Cordyceps chanhua*.

As you are concerned, *Cordyceps cicadae* should be *Cordyceps chanhua*. The reason is that Chinese scholar Li Zengzhi found the sexual *Cordyceps chanhua*,

classified it as a new species of the genus *Cordyceps*, and named it *Cordyceps chanhua* with the ancient name of *Cordyceps cicadae* (1).

REFERENCE

1. Li ZZ, Luan FG, Nigel HJ, Zhang SL, Chen MJ, Huang B, Shun CS, Chen ZA, Li CR, Tan YJ, Dong JF. Biodiversity of cordycipitoid fungi associated with *Isaria cicadae* Miquel II: Teleomorph discovery and nomenclature of chanhua, an important medicinal fungus in China. *Mycosystema*, 2021, 40(2): 1-12. <https://doi.org/10.13346/j.mycosystema.200119>.

Thank you again for your careful checks. Based on your comments, we have corrected some other minor errors in the manuscript.

Reviewer #2 (Comments for the Author):

In this manuscript, the authors analyzed the bacterial community in different parts of C. cicadae in different production stages and different environments by high-throughput sequencing. They also demonstrated that the filtration effect of C. cicadae membrane on bacterial communities. Furthermore, they also found the nitrogen transport function of the membrane of C.cicadae. The obtained information provide a basis for understanding the mechanism of the internal microbial community maintained by C. cicadae and nutritional balance by self-regulation in C. cicadae.

Specific comments and suggestions:

1. L68-71, it is confused for the researches on the morphological characteristics of the mycoderm in the sentence "most studies on the morphological characteristics of cordyceps fungi focus on the mycoderm, sclerotia, stroma and ascus structures of cordyceps (24, 25, 26). However, as a special structure of some entomopathogenic fungi, the study on the morphological characteristics of the mycoderm has not been reported." The authors should explain and revise.

Response: Thanks for your careful checks. We are sorry for our carelessness. Based on your comments, we have made the corrections to make the word harmonized within the whole manuscript.

2. L206, the subtitle for "Apparent Morphology of C. cicadae Mycoderm in Soil Atmosphere", it should be "Soil and Atmosphere".

Response: We have made the corresponding changes in the manuscript.

3. L240-248, It is better to provide the morphology features of CFU (or at least several representative images for the morphology character of CFU).

Response: We have included pictures of CFU morphological characteristics of soil bacteria cultured on plate medium in the supplementary material.

4. L497, "at the concentration of 500 mL", it is not concentration.

Response: Thanks for your careful checks. Based on your comments, we have made the corresponding changes in the manuscript.

5. L538-539, The details for equipment and parameter should be provided.

Response: We appreciate your valuable comments. We have added the specific equipment parameters of L538-539 to the corresponding positions in the manuscript, marked with red font

6. L863-866 What did the authors mean for the indicated arrow, the author should indicate.

Response: The arrows in the picture represent the "H" type fusion hyphae partial zoom. We have highlighted it in the notes.

7. L881, the description should be used by English.

Response: Thanks for your careful checks. We have made the corresponding changes in the manuscript.

8. It should be "discussion" (Line 295).

Response: Thanks for your careful checks. We have made the corresponding changes in the manuscript.

October 22, 2023

Dr. Xiao Zou
Guizhou University
College of Life Sciences
Guiyang
China

Re: Spectrum01179-23R3 (Filtration Effect of *Cordyceps chanhua* Mycoderm on Bacteria and its Transport Function on Nitrogen)

Dear Dr. Xiao Zou:

Thank you for submitting your manuscript to Microbiology Spectrum. However, instead of mentioning the changes made in the Response file, the manuscript has NOT been corrected and updated in re-ordering figures and Tables etc. This kind of mistakes is unthinkable. Multiple rounds of revisions may not be permitted.

Link Not Available

Sincerely,

Chengshu Wang

Journals Department
Reviewer comments:

Instead of mentioning the changes made in the Response file, the manuscript has NOT been corrected and updated in re-ordering figures and Tables etc. This kind of mistakes is unthinkable. Multiple rounds of revisions may not be permitted.

Staff Comments:

Preparing Revision Guidelines

Please return the manuscript within 60 days; if you cannot complete the modification within this time period, please contact me. If you do not wish to modify the manuscript and prefer to submit it to another journal, please notify me of your decision immediately so that the manuscript may be formally withdrawn from consideration by Microbiology Spectrum.

Response to Reviewers

We sincerely thank the editor and all reviewers for their valuable feedback that we have used to improve the quality of our manuscript. The reviewer comments are laid out below in italicized font and specific concerns have been numbered. Our response is given in normal font and changes/additions to the manuscript are given in the red text.

Response to the reviewer on October 14

1. *Cordyceps cicadae* is still used in Supplementary.

Response: Thank you for your careful checks. And we've already changed *Cordyceps cicadae* to *Cordyceps chanhua* in the Supplementary.

2. Merge Fig. 8 and Fig. 9 into a figure with two panels.

Response: We have merged Fig. 8 and Fig. 9 into a figure with two panels, and the new number is Fig. 8 in the manuscript. The corresponding figures and legends have been placed after the references in the manuscript. The final result is shown below.

Fig. 8 $\delta^{15}\text{N}$ values in different samples in experiment D and experiment E. A: $\delta^{15}\text{N}$ flows from *Cordyceps chanhua* to soil samples. B: $\delta^{15}\text{N}$ flows from soil to *Cordyceps chanhua* samples.

3. Move the Figures 10-13 mentioned in Methods to Supplementary and reorder them.

Response: Thanks for your advice and we have moved the Figures 10-13 mentioned in Methods to Supplementary and re-order them. And Figures 10-13 Corresponding numbers in supplementary are shown in Fig. S6-S9.

4. It is not appropriate to have quite a few (small) Tables, Table 8 should go to supplementary. What are the relationships regarding Table 6 vs. Fig. 7, and Table 7 vs. Fig. 9, the same things? If being similar, please move Tables 6 and 7 to supplementary as well.

Response: Thanks for your careful checks. We have moved Table 8 into the supplementary, and the corresponding number in the supplementary material is Table S3. In addition, Table 6 and Table 7 show the ^{15}N and total nitrogen content of different samples in experiment D and experiment E, respectively. Fig. 8 and Fig. 9 (Now merged into Fig. 8) respectively show the enrichment degree of each sample of ^{15}N in experiment D and experiment E. We have now moved Table 6 and Table 7 into the supplementary, and the corresponding number in the supplementary material is Table S1 and Table S2.

Reviewer #1 (Comments for the Author)

It's not easy to understand the experimental and methodological methods. Maybe some figures or schematic diagrams should be added to make it clear, such as the filtration effect and N15 flow experiments. And also, the text is not set out well.

Response: We feel great thanks for your professional review work on our article. As you are concerned, there are several problems that need to be addressed. According to your nice suggestions, we have made corrections to our previous draft, the detailed corrections are listed below.

(1) We have corrected some textual errors in the article, and the changes to the manuscript have been shown in red text.

(2) In order to better understand the experiment and methodological methods, we drew some schematic diagrams in the manuscript. We have put schematic diagrams in the corresponding position in the manuscript.

Response to the reviewer on October 10

1. The Microbial Diversity is the diversity of mycoderm? Please specified throughout the text.

We apologize for not specifying the meaning of microbial diversity in the text, and the microbial diversity in this study refers to the bacterial community diversity of internal sclerotium and mycoderm of *Cordyceps Chanhua* at different growth stages and under different cultivation environments, we have specified in the manuscript.

2. Why there is no results of fungal diversity.

In manuscript L71~73, previous studies have found that the relative abundance of other fungi in the sclerotia of *C. chanhua* was relatively low, while the bacterial community was relatively rich, therefore, we didn't study fungal diversity.

3. If the pupae of *Antheraea pernyi* is sterilized thoroughly before being infection, *C. cicadae* can develop normally? It should be discussed.

Thank you very much for your valuable advice on our manuscript, to better answer this question, we designed an experiment on the effect of complete disinfection of the pupae of *Antheraea pernyi* on the growth and development of *C. chanhua*. We designed a total of 4 groups of experiments, namely (1) disinfection was carried out according to the disinfection method of the pupae of *Antheraea pernyi* in the original manuscript (d₁): wash the surface of *A. pernyi* with tap water and dry it naturally. Soak in 75% alcohol in the sterile super clean table and scrub the insect surface with cotton for about ten seconds, then rinse it with sterile water, clip it into a white porcelain plate covered with sterile paper towels, and dry the surface water; (2) the sodium hypochlorite solution was added on the basis of the experimental group d₁

(d₂): First rinse the surface of the surface of *A. pernyi* with tap water, dry naturally; the insect body surface was soaked with 75% alcohol and scrubbed with cotton for about 10 s in a sterile ultra-clean table, and then washed with 5% sodium hypochlorite solution for 10 s. The most sterile water was used for washing, and then placed on a white porcelain plate covered with sterile paper towels to absorb the surface moisture; (3) a slight improvement was made on the basis of experimental group d₂ (d₃): First rinse the surface of the surface of *A. pernyi* with tap water, dry naturally; the insect body surface was soaked with 75% alcohol and scrubbed with cotton for about 20 s in a sterile ultra-clean table, and then washed with 5% sodium hypochlorite solution for 20 s. The most sterile water was used for washing, and then placed on a white porcelain plate covered with sterile paper towels to absorb the surface moisture; (4) the surface of *A. pernyi* was completely sterilized at 121 °C for 30 min (d₄). The growth and development of *C. Chanhua* 25 days after inoculation were shown in Table 1.

Table 1 Growth information of *Cordyceps Chanhua* under different disinfection methods of the pupae of *Antheraea pernyi*

Method of disinfection	Total pupae	Number of fossilized pupae	Rate of rigidity	Number of fruiting bodies
d ₁	12	11	91.6%	8.66667±1.08012
d ₂	12	12	100.00%	8.88889±0.69611
d ₃	12	11	91.67%	8.77778±0.77778
d ₄	12	3	25.00%	5.00000±0.57735

As can be seen from Table 1, 25 days after inoculation, d₂ disinfection treatment had the highest rate of *C. Chanhua* rigidor and the highest number of fruite bodies, however, d₄ had the lowest ossification rate and number of fruites. Therefore, the pupae of *A. pernyi* sterilization is not suitable for *C. Chanhua* cultivation.

4. *Cordyceps cicadae* should be *Cordyceps chanhua*.

As you are concerned, *Cordyceps cicadae* should be *Cordyceps chanhua*. The reason is that Chinese scholar Li Zengzhi found the sexual *Cordyceps chanhua*,

classified it as a new species of the genus *Cordyceps*, and named it *Cordyceps chanhua* with the ancient name of *Cordyceps cicadae* (1).

REFERENCE

1. Li ZZ, Luan FG, Nigel HJ, Zhang SL, Chen MJ, Huang B, Shun CS, Chen ZA, Li CR, Tan YJ, Dong JF. Biodiversity of cordycipitoid fungi associated with *Isaria cicadae* Miquel II: Teleomorph discovery and nomenclature of chanhua, an important medicinal fungus in China. *Mycosystema*, 2021, 40(2): 1-12. <https://doi.org/10.13346/j.mycosystema.200119>.

Thank you again for your careful checks. Based on your comments, we have corrected some other minor errors in the manuscript.

Reviewer #2 (Comments for the Author):

In this manuscript, the authors analyzed the bacterial community in different parts of C. cicadae in different production stages and different environments by high-throughput sequencing. They also demonstrated that the filtration effect of C. cicadae membrane on bacterial communities. Furthermore, they also found the nitrogen transport function of the membrane of C.cicadae. The obtained information provide a basis for understanding the mechanism of the internal microbial community maintained by C. cicadae and nutritional balance by self-regulation in C. cicadae.

Specific comments and suggestions:

1. L68-71, it is confused for the researches on the morphological characteristics of the mycoderm in the sentence "most studies on the morphological characteristics of cordyceps fungi focus on the mycoderm, sclerotia, stroma and ascus structures of cordyceps (24, 25, 26). However, as a special structure of some entomopathogenic fungi, the study on the morphological characteristics of the mycoderm has not been reported." The authors should explain and revise.

Response: Thanks for your careful checks. We are sorry for our carelessness. Based on your comments, we have made the corrections to make the word harmonized within the whole manuscript.

2. L206, the subtitle for "Apparent Morphology of *C. cicadae* Mycoderm in Soil Atmosphere", it should be "Soil and Atmosphere".

Response: We have made the corresponding changes in the manuscript.

3. L240-248, It is better to provide the morphology features of CFU (or at least several representative images for the morphology character of CFU).

Response: We have included pictures of CFU morphological characteristics of soil bacteria cultured on plate medium in the supplementary material.

4. L497, "at the concentration of 500 mL", it is not concentration.

Response: Thanks for your careful checks. Based on your comments, we have made the corresponding changes in the manuscript.

5. L538-539, The details for equipment and parameter should be provided.

Response: We appreciate your valuable comments. We have added the specific equipment parameters of L538-539 to the corresponding positions in the manuscript, marked with red font

6. L863-866 What did the authors mean for the indicated arrow, the author should indicate.

Response: The arrows in the picture represent the "H" type fusion hyphae partial zoom. We have highlighted it in the notes.

7. L881, the description should be used by English.

Response: Thanks for your careful checks. We have made the corresponding changes in the manuscript.

8. It should be "discussion" (Line 295).

Response: Thanks for your careful checks. We have made the corresponding changes in the manuscript.

9. You still need to include an in-text callout for figure 9 in the text.

Response: Thanks a lot. We have merged Fig. 8 and Fig. 9 into a figure with two panels, and the new number is Fig. 8 in the manuscript. That means there are only eight figures left in our article. In addition, we have added in-text callout to all the figures in the manuscript. And we have moved the Figures 10-13 mentioned in Methods to Supplementary and re-order them. And Figures 10-13 Corresponding numbers in supplementary are shown in Fig. S6-S9.

Re: Spectrum01179-23R4 (Filtration Effect of *Cordyceps chanhua* Mycoderm on Bacteria and its Transport Function on Nitrogen)

Dear Dr. Xiao Zou:

Unfortunately, after multiple rounds of modifications, the authors still failed to correct the supplementary figures in this fourth revision. For example, Figures 10-14 are still used in Methods.

Your manuscript has been provisionally accepted, and I am forwarding it to the ASM production staff for publication. Your paper will first be checked to make sure all elements meet the technical requirements. ASM staff will contact you if anything needs to be revised before copyediting and production can begin. Otherwise, you will be notified when your proofs are ready to be viewed.

Sincerely,
Chengshu Wang
Editor
Microbiology Spectrum